# Liver sinusoidal endothelial cells constitute a major route for hemoglobin clearance

Gabriela Zurawska[1,9], Zuzanna Sas[2,9], Aneta Jończy[1,9], Raghunandan Mahadeva [1], Patryk Slusarczyk[1], Marta Chwałek[1], Daniel Seehofer[3], Georg Damm[3], Rafał Mazgaj [4], Marcin Skórzyński[5], Maria Kulecka[6], Izabela Rumieńczyk[6], Morgane Moulin[7], Kamil Jastrzębski[1], Kevin Waldron [4], Michal Mikula[6], Anders Etzerodt[7], Remigiusz Serwa [8], Marta Miączyńska [1], Tomasz P Rygiel [5,10]✉ & Katarzyna Mleczko-Sanecka [1,10]✉

## Abstract

**Mild rupture of aged erythrocytes occurs in the spleen, resulting in hemoglobin (Hb) release, whereas pathological hemolysis characterizes several diseases. Hb detoxification is attributed to macrophages, but other routes of Hb clearance remain elusive. Here, we uncover that Hb uptake is chiefly executed by liver sinusoidal endothelial cells (LSECs) via macropinocytosis. Consistently, LSECs display proteomic signatures indicative of heme catabolism, ferritin iron storage, antioxidant defense, and macropinocytic capacity, alongside high iron content and expression of the iron exporter ferroportin. Erythrocyte/Hb transfusion assays demonstrate that splenic macrophages excel in erythrophagocytosis, while LSECs and Kupffer cells scavenge the spleen-borne hemolysis products Hb and erythrocyte membranes, respectively. High Hb doses result in transient hepatic iron retention, LSEC-specific induction of heme-catabolizing *Hmox1*, along with the iron-sensing *Bmp6*-hepcidin axis culminating in hypoferremia. Transcriptional induction of *Bmp6* in LSECs is phenocopied by erythrocyte lysis upon phenylhydrazine and elicits a distinct transcriptional signature compared to iron. Collectively, we identify LSECs as key Hb scavengers, a function that establishes the spleen-to-liver axis for iron recycling and contributes to heme detoxification during hemolysis.**

**Keywords** Hemoglobin; Hemolytic Disease; LSEC; Macropinocytosis; RBC
**Subject Categories** Autophagy & Cell Death; Haematology; Immunology

## Introduction

Internal iron recycling from senescent red blood cells (RBCs) satisfies most of the body's iron needs (Muckenthaler et al, 2017; Slusarczyk and Mleczko-Sanecka, 2021) and relies mainly on the phagocytic clearance of aged RBCs by splenic red pulp macrophages (RPMs) (Ma et al, 2021; Youssef et al, 2018). However, recent evidence suggests that some aged RBCs escape erythrophagocytosis and lyse locally in the spleen, thus releasing hemoglobin (Hb) (Klei et al, 2020). Several inherited and acquired disorders, including hereditary anemias, autoimmune hemolytic diseases, or infections, are characterized by compromised erythrocyte stability and an increased risk of hemolysis (Kato et al, 2017; Schaer et al, 2014). Free Hb is captured by the acute-phase plasma protein haptoglobin (Wang et al, 2001). The Hb-haptoglobin complexes are sequestered via CD163 (Kristiansen et al, 2001), a receptor that is highly expressed by both splenic RPMs and liver macrophages, Kupffer cells (KCs) (Klei et al, 2020; Slusarczyk and Mleczko-Sanecka, 2021). However, the role of these macrophage populations in Hb uptake has not been well characterized. Interestingly, pharmacokinetic studies in non-rodent mammals have shown that the clearance rate of the Hb-haptoglobin complex is significantly slower than that of free Hb (Boretti et al, 2014), and that Hb sequestration may occur independently of haptoglobin and/or CD163 (Etzerodt et al, 2013; Schaer et al, 2006b). These observations are clinically relevant as enhanced erythrophagocytosis and prolonged erythrolytic conditions lead to the partial loss of the CD163-expressing iron-recycling macrophages (Theurl et al, 2016; Youssef et al, 2018) and are characterized by depletion of the plasma haptoglobin pool (Schaer et al, 2014; Vinchi et al, 2013; Vinchi et al, 2021). Collectively, this evidence suggests that macrophages may be dispensable for Hb clearance. It is well established that free Hb undergoes renal glomerular filtration (Fagoonee et al, 2005; Marro et al, 2007). However, it remains

[1]International Institute of Molecular and Cell Biology in Warsaw, Warsaw, Poland. [2]Medical University of Warsaw, Warsaw, Poland. [3]Department of Hepatobiliary Surgery and Visceral Transplantation, Clinic for Visceral, Transplant, Thoracic and Vascular Surgery, Leipzig University Medical Center, Leipzig, Germany. [4]Institute of Biochemistry and Biophysics, Polish Academy of Sciences, Warsaw, Poland. [5]Mossakowski Medical Research Institute, Polish Academy of Sciences, Warsaw, Poland. [6]Maria Sklodowska-Curie National Research Institute of Oncology, Warsaw, Poland. [7]Department of Biomedicine, Aarhus University, Aarhus, Denmark. [8]ReMedy International Research Agenda Unit, IMol Polish Academy of Sciences, Warsaw, Poland. [9]These authors contributed equally: Gabriela Zurawska, Zuzanna Sas, Aneta Jończy. [10]These authors contributed equally: Tomasz P Rygiel, Katarzyna Mleczko-Sanecka. ✉E-mail: trygiel@imdik.pan.pl; kmsanecka@iimcb.gov.pl

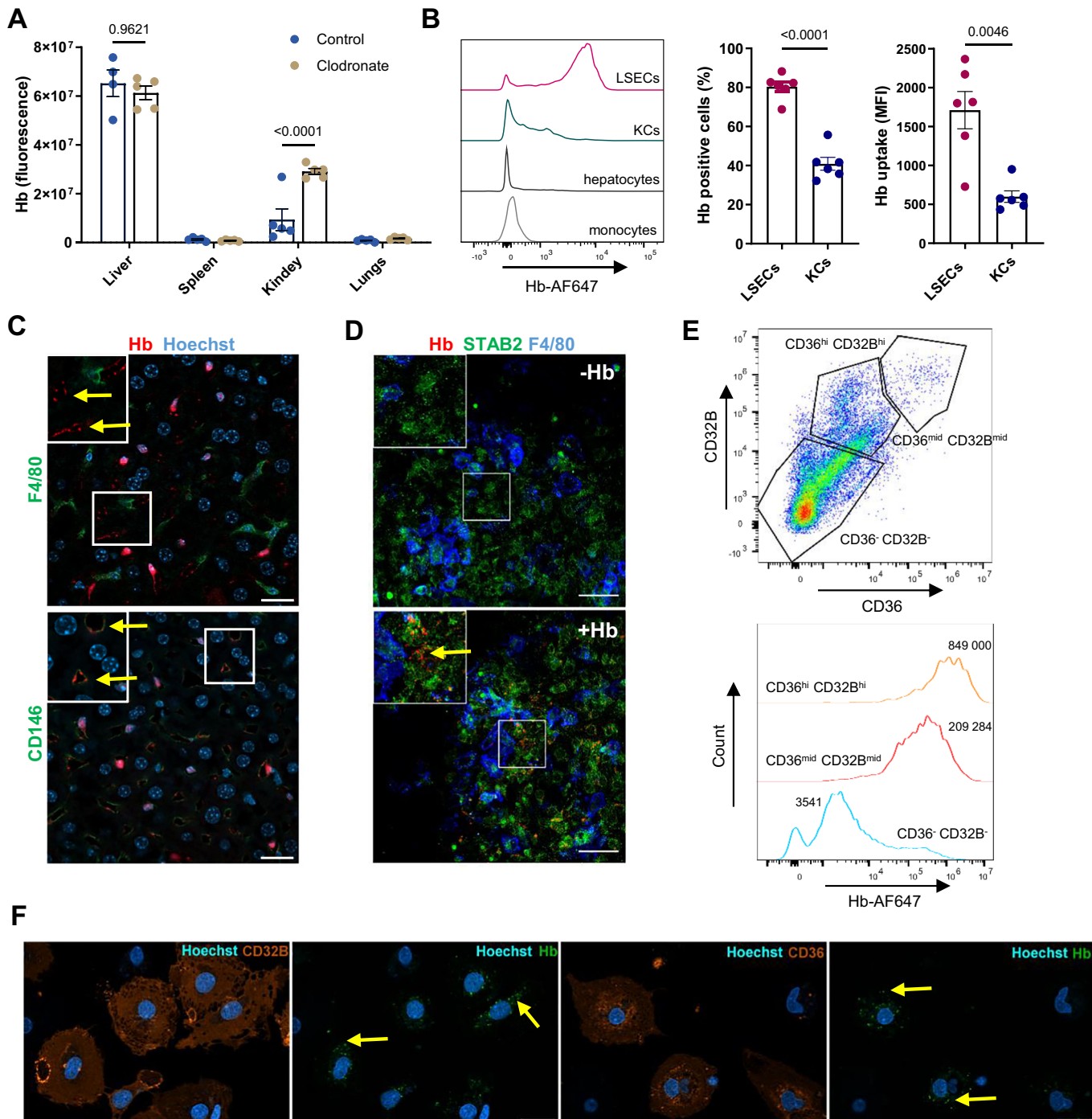

unclear whether other specialized routes of extra-renal and macrophage-independent Hb clearance operate in the body.

The liver receives approximately 25% of the cardiac output and is exposed to blood from the portal circulation, which drains the gastrointestinal tract and the spleen (Poisson et al, 2017). The hepatic capillary network, which is composed of venous sinusoids, is specialized for monitoring and filtering blood components. Liver sinusoidal endothelial cells (LSECs), along with KCs, constitute the most efficient dual scavenging system in the body (Sorensen et al, 2012). While KCs engulf large particles, LSECs remove

macromolecules and nanoparticles, a function that protects the body from waste by-products and noxious blood factors (Koch et al, 2021; Schledzewski et al, 2011). The maintenance of appropriate iron homeostasis adds to the growing spectrum of homeostatic functions of LSECs. Importantly, LSECs are the sensors of body iron levels and the major producers of inducible angiokine bone morphogenetic protein 6 (BMP6), and homeostatic BMP2 (Canali et al, 2017; Koch et al, 2017). BMPs function as upstream activators of hepcidin, a key iron-regulatory hormone produced by liver hepatocytes (Muckenthaler et al, 2017). Hepcidin

**Figure 1. LSECs represent the major cell type that sequesters Hb.**

(A) Hemoglobin (Hb) distribution in control and macrophage-depleted mice (clodronate) injected with AlexaFluor 750-labeled Hb (Hb-AF750, 10 μg/mouse), imaged with Bruker in vivo Imaging System. Results are presented as total fluorescent counts from the isolated organs. (B) AF750 fluorescence from the indicated liver cell populations isolated from Hb-AF750-injected mice was analyzed with flow cytometry. (C) Frozen liver slices from mice injected with Hb-AF647 (red) were processed and stained for Kupffer cells (KCs) (F4/80, green) or LSECs (CD146, green) and nuclei (blue). Single-channel images for this figure are presented in Fig. EV1C. (D) Murine liver primary non-parenchymal cells (NPCs) were treated in vitro with Hb-AF750 (red) (0.5 μg/ml) and stained for F4/80 (blue) and the LSEC marker STAB2 (green). (C, D) Areas in the highlighted rectangles are shown at 5× higher magnification. (E) Flow cytometric analysis of human freshly isolated primary NPCs, exposed in culture to human Hb-AF647 and stained with the human LSEC markers CD36 and CD32B. The intensity of Hb-A647 uptake was measured in two LSEC subpopulations, CD36$^{hi}$CD32B$^{hi}$ and CD36$^{mid}$CD32B$^{mid}$, as compared to other NPCs CD36$^-$CD32B$^-$. Numbers indicate MFI. Data are representative of three independent donors of human NPCs. (F) Hb-AF647 (green) uptake was imaged in freshly isolated human liver NPCs stained for the LSEC markers CD36 and CD32B (orange). (C, D, F) Arrows indicate Hb presence in LSECs; Scale bars, 20 μm. Numerical data are expressed as mean ± SEM; each data point represents one biological replicate (individual mouse or independent cell-based experiment), n = 4–5 (A), 6 (B). Two-way ANOVA with Tukey's Multiple Comparison tests was used in (A), and Welch's unpaired t test was used to determine statistical significance in (B); exact P values are shown on graphs. Source data are available online for this figure.

suppresses iron release into the bloodstream via the sole known iron exporter ferroportin (FPN), thereby limiting iron availability under iron-rich conditions (Mleczko-Sanecka and Silvestri, 2021). However, it remains unclear how different types of iron signals are sensed and detoxified in the liver microenvironment, and whether the emerging scavenging functions of LSECs cross-talk with their role in maintaining iron homeostasis. Here, we show that LSECs constitute a major route for Hb clearance. They contribute to steady-state iron recycling from spleen-derived Hb that enters the liver via the portal vein and participate in heme detoxification during hemolysis, timely coupled with induction of the iron-sensing BMP6–hepcidin axis.

## Results

### LSECs engage macropinocytosis to sequester Hb

Studies using radiolabeled Hb have shown that the clearance of injected hemoglobin is rapid (Fagoonee et al, 2005) and mainly mediated by the liver, kidney, and spleen (Fagoonee et al, 2005; Marro et al, 2007). To address the involvement of macrophages in Hb uptake, we first performed experiments using clodronate liposomes, a well-established strategy for depleting tissue macrophages, including hepatic KCs. Using a whole-organ imaging system, we observed that regardless of the presence of KCs, the liver emerged as the major organ sequestering fluorescently-labeled mouse Hb (Figs. 1A and EV1A,B). Next, using flow cytometry and confocal microscopy imaging of mouse liver sections, we identified CD146$^+$STAB2$^+$ LSECs as the major hepatic cell type that takes up Hb, surpassing KCs, hepatocytes, and monocytes (Figs. 1B,C and EV1C; see Appendix for gating strategies). We found that murine primary LSEC cultures recapitulated the high capacity of this cell type for Hb uptake, as validated by confocal microscopy and flow cytometry (Figs. 1D and EV1D). We further confirmed that freshly isolated human primary LSECs, distinguished by CD36 and CD32B markers, as previously reported (Ganesan et al, 2012; Strauss et al, 2017) and confirmed by the Human Liver Cell Atlas (Guilliams et al, 2022) (Fig. EV1E), outperformed other NPCs in human Hb uptake (Fig. 1E,F). Notably, flow cytometric analysis revealed that cells with high CD36 and CD32B expression, albeit not highly abundant, showed the highest capacity for Hb scavenging (Fig. 1E).

To better understand the mechanism of Hb uptake, we used mouse primary liver cell cultures. In contrast to murine KCs, LSECs were negative for the Hb-haptoglobin complex receptor, CD163 (Fig. 2A), and took up free Hb to a comparable extent as haptoglobin-bound Hb (Hb:Hp; Fig. 2B). Therefore, we hypothesized that LSECs may sequester Hb in a receptor-independent manner, possibly via macropinocytosis, which is a non-specific internalization of extracellular fluid (Canton, 2018). Staining of primary murine LSECs with an early endosome marker, EEA1, revealed many constitutively formed intracellular vesicles of macropinosome size (1–2 μm) (Figs. 2C and EV2A). Fluorescently-labeled Hb was entrapped, though not exclusively, within such vesicles (Figs. 2C and EV2A), and co-localized with high-molecular-weight dextran, a known macropinocytic cargo (Figs. 2D and EV2B) (Commisso et al, 2013; Zdzalik-Bielecka et al, 2021). Human primary CD36-high LSECs likewise displayed numerous macropinosome-like vesicles and efficiently sequestered dextran (Fig. 2E,F). Consistently, ethyl-isopropyl amiloride (EIPA), a well-established blocker of macropinocytosis, in contrast to an inhibitor of clathrin-mediated endocytosis, chlorpromazine (CPZ), abolished Hb uptake in murine primary LSECs (Figs. 2G and EV2C) and decreased Hb scavenging by their human counterparts (Fig. 2H). Notably, EIPA was less potent in inhibiting Hb uptake in human LSECs, indicating possible species differences in sensitivity or compensatory uptake pathways. Furthermore, Hb internalization by murine LSECs was partially dependent on actin remodeling (suppressed by latrunculin A, LAT-A), and Cdc42 activity (inhibited by ML141), both of which are important for macropinosome formation (Zdzalik-Bielecka et al, 2021). Hb uptake also required canonical Wnt signaling via β-catenin (blocked by PRI-724), a pathway that is active in LSECs (Klein et al, 2008) and has previously been shown to activate macropinocytosis (Fig. 2G) (Redelman-Sidi et al, 2018; Tejeda-Munoz et al, 2019). Similarly, Hb uptake by human primary LSECs was suppressed by LAT-A (Fig. 2H), supporting the conclusion that Hb uptake in LSECs is at least partially dependent on macropinocytosis rather than classical clathrin-mediated endocytosis, even though the degree of sensitivity to EIPA may differ between species. Noteworthy, Hb uptake was not altered by the micropinocytosis blocker, nystatin (Fig. 2G). Interestingly, close to 100% of LSECs actively sequestered Hb even at low doses, and the capacity for the uptake did not show saturation with higher cargo presence, implying a receptor-independent entry route (Fig. EV2D). However, the same doses

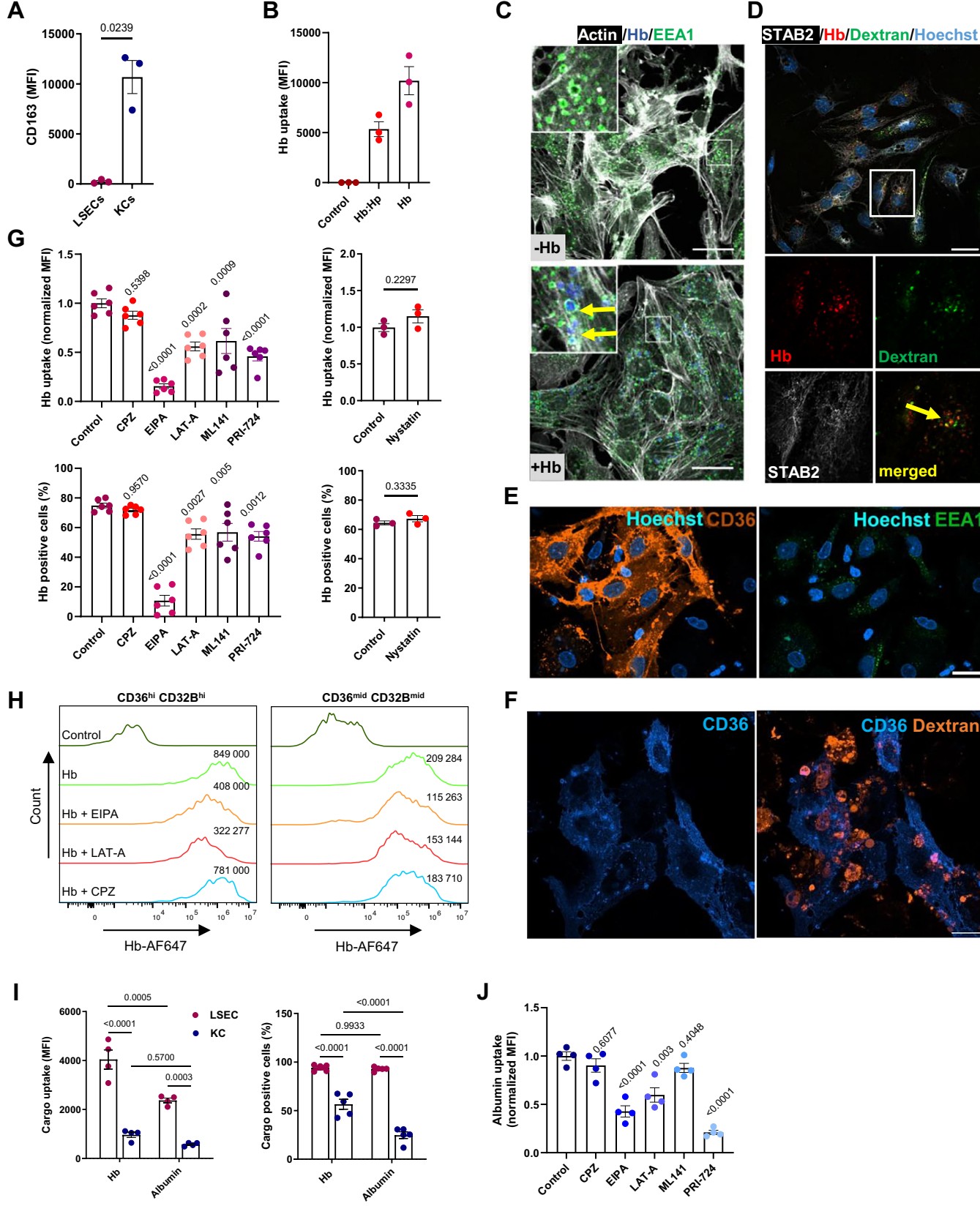

◀ **Figure 2. LSECs engage macropinocytosis to sequester Hb.**

(A) Cell-surface expression of the CD163 receptor on LSECs and KCs was measured by flow cytometry and presented as fluorescence intensity. (B) Uptake of free Hb or Hb in the presence of haptoglobin (Hp, Hb:Hp) in primary murine LSEC cultures was measured with flow cytometry. (C) Hb-AF647 vesicle localization was imaged with microscopy in NPCs in vitro cultures depleted of macrophages. Arrows indicate Hb-AF647 (blue) presence in the EEA1$^+$ (green) and Actin$^+$ (white) vesicles. Areas in the highlighted rectangles are shown at a higher 5× magnification (left corner). Scale bars, 20 μm. Single-channel images for this figure are presented in Fig. EV2A. (D) Co-localization of Hb-AF647 (red) with rhodamine-dextran (green) was imaged with microscopy in STAB2$^+$ LSECs (white); nuclei are stained in blue. The arrow indicates colocalization of Hb and rhodamine-dextran. The area in the highlighted rectangle is shown at a higher 5× magnification in separate channels below. Scale bar, 20 μm. Single-channel images for this figure are presented in Fig. EV2B. (E) EEA1$^+$ vesicles (green) were imaged with microscopy in human NPCs cultures stained with the LSEC marker CD36 (orange); nuclei are stained in blue. (F) Texas red-dextran uptake (orange) was imaged with microscopy in human NPCs cultures stained with the LSEC marker CD36 (blue). (G) NPCs were pre-treated with the clathrin-mediated endocytosis inhibitor chlorpromazine (CPZ), the macropinocytosis blocker EIPA, actin polymerization blocker latrunculin A (LAT-A), inhibitor of Cdc42 GTPase ML141, β-catenin inhibitor PRI-724, and with micropinocytosis blocker nystatin before Hb-AF750 treatment for 10 min. Flow cytometry was used to determine Hb uptake by the LSECs. (H) Flow cytometric analysis of CD36$^{hi}$CD32B$^{hi}$ and CD36$^{mid}$CD32B$^{mid}$ human primary LSECs, treated with Hb-AF647 and exposed to EIPA, LAT-1, and CPZ. Numbers indicate MFI. Data are representative of three independent donors of human NPCs. (I) Mice were injected for 1 h with Hb-AF750 or albumin-AF750, both at 10 μg/mouse. The fluorescence intensity of AF750, the percentage of AF750$^+$ LSECs, and KCs populations were measured with flow cytometry. (J) NPCs were pre-treated with CPZ, EIPA, LAT-A, ML141, and PRI-724 before albumin-AF647 treatment for 10 min. Flow cytometry was used to determine albumin uptake by the LSECs. (C–F) Scale bars, 20 μm. Numerical data are expressed as mean ± SEM, each data point represents one biological replicate (individual mouse or independent cell-based experiment), $n = 3$ (A, B), 3–6 (G), 4–5 (I), 4 (J). Welch's unpaired *t* test was used to determine statistical significance in (A, G) for nystatin, two-way ANOVA with Tukey's Multiple Comparison tests was used in (I); while one-way ANOVA with Tukey's Multiple Comparison tests was used in (G, J). Exact *P* values are shown on graphs; values above bars in (G, J) indicate comparison with untreated controls. Source data are available online for this figure.

of dextran, a cargo of similar molecular weight, were internalized by LSECs with far lower efficiency (Fig. EV2D). This implied a certain specialization of LSECs towards Hb scavenging, even though the uptake of dextran by LSECs was likewise abolished by the macropinocytosis inhibitor EIPA or actin remodeling blocker (Fig. EV2E). Confirming the high macropinocytic capacity of LSECs in vivo, we demonstrated their ability for efficient uptake of fluorescently-labeled albumin, another known macropinocytic cargo (Fig. 2I) (Commisso et al, 2013). Furthermore, albumin clearance by primary LSECs was inhibited by EIPA, LAT-A, and PRI-724, further supporting the high macropinocytic activity of these specialized cells (Fig. 2J). Of note, we observed that the uptake of Hb by primary murine KCs was not affected by the clathrin endocytosis inhibitor CPZ, and appeared to be dependent on macropinocytosis (Fig. EV2F). Interestingly, the uptake of the Hb:Hp complex by LSECs and KCs was suppressed by both EIPA and CPZ, implying the involvement of both clathrin-dependent endocytosis and macropinocytosis routes in this process (Fig. EV2G). Taken together, our data identify LSECs as efficient scavengers of both free and haptoglobin-bound Hb and implicate macropinocytosis as an important alternative pathway for Hb uptake, which also extends to KC-mediated Hb scavenging.

## LSECs outperform other endothelial and macrophage populations in Hb uptake

Similar to the liver, sinusoidal endothelial cells are also present in the spleen and bone marrow (Koch et al, 2021), organs that perform critical functions in systemic iron metabolism owing to the presence of CD163$^+$ iron-recycling RPMs and erythroblastic island macrophages, respectively. Therefore, we sought to accurately compare the Hb-scavenging capacity of endothelial and macrophage populations from the spleen and bone marrow with those of hepatic LSECs and KCs over a wide range of Hb doses. Thus, we intravenously injected mice with 1 μg of Hb, which is below the binding capacity of circulating haptoglobin (10–20 μg/ml plasma), an intermediate dose of 10 μg, and 100 μg (injected together with 10 mg of unlabeled Hb), which saturates the haptoglobin pool and

mimics hemolytic conditions. We also injected mice with 1 μg of Hb bound to haptoglobin. Myeloid cells and ECs from the bone marrow were not capable of efficient Hb sequestration (Appendix Fig. S1A). Strikingly, we observed the extraordinary ability of LSECs to internalize Hb as compared to the other cell types analyzed, as exemplified by the 70–99% of Hb-positive cells depending on the dose (Fig. 3A) and the highest mean intensity of the Hb signal per cell as compared to KCs, splenic ECs, or RPMs (Fig. 3B,D,F,H). The scavenging capacity of KCs and splenic ECs gradually increased with the Hb dose (Fig. 3C–F), whereas RPMs contributed, albeit slightly, to Hb clearance only at the highest dose, mimicking hemolysis (Fig. 3G,H). Finally, using flow cytometry and whole-organ imaging, we did not detect Hb accumulation in the aorta, lined by non-sinusoidal ECs (Appendix Fig. S1B,C). In conclusion, our data suggest that LSECs outperform other cell types for Hb clearance over a wide range of circulating Hb concentrations.

## LSECs constitutively express proteins critical for iron recycling from heme

We were intrigued by the fact that LSECs efficiently scavenged low doses of Hb. To further explore their specialization, we examined data from single-cell RNA sequencing of ECs from different mouse organs (Kalucka et al, 2020). Interestingly, both *Hmox1* [encoding heme oxygenase 1 (HO-1)] and *Blrvb* (encoding biliverdin reductase b, BLRVB), which are important for heme catabolism, together with important antioxidant enzymes such as *Gclc*, *Gpx4*, and *Nqo1*, were identified by Kalucka et al (Kalucka et al, 2020) as metabolic markers exclusively specific for liver EC. Using the EC Atlas, we visualized the expression levels of *Hmox1* and *Blvrb*, as well as *Slc40a1*, the gene encoding the iron exporter FPN, and found a clear signature for the high expression of these transcripts in liver ECs compared to other organs (Appendix Fig. S2A). Consistently, by employing dedicated single-cell RNA-seq databases, we observed high expression levels of *Slc40a1*, *Hmox1*, and *Hmox2* in pig liver ECs and *Slc40a1* in human LSECs (Appendix Fig. S2B,C) (Guilliams et al, 2022; Wang et al, 2022). Following up

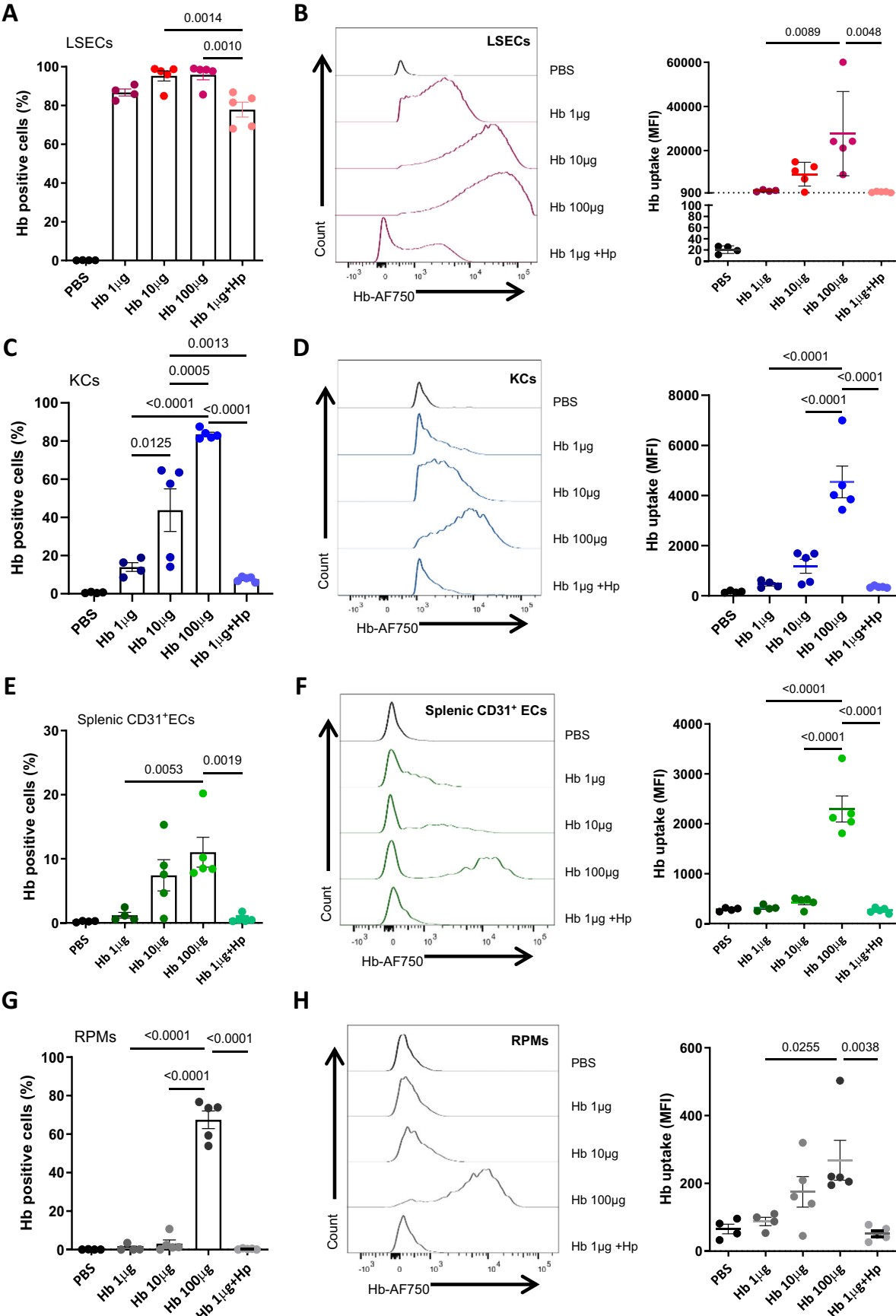

◀ **Figure 3. LSECs outperform other endothelial populations and macrophages in Hb uptake.**

Mice received intravenous injections of fluorescently labeled hemoglobin (Hb-AF750) at increasing doses, either alone or complexed with haptoglobin, as indicated. Samples were collected after 1h. The percentage of Hb-AF750-positive cells and fluorescence intensity of Hb-AF750 are shown for LSECs (**A, B**), KCs (**C, D**), splenic CD31$^+$ endothelial cells (ECs) (**E, F**), and RPMs (**G, H**). Both parameters were analyzed with flow cytometry. Data are expressed as mean ± SEM, and each data point represents one biological replicate (individual mouse), $n = 4$–5 (**A–H**). One-way ANOVA with Tukey's Multiple Comparison tests was used to determine statistical significance; exact *P* values are shown on graphs. Source data are available online for this figure.

on these transcriptomic signatures, we detected protein expression of FPN and HO-1 in mouse liver sections (Fig. 4A) and primary murine LSECs (Fig. 4B). Flow cytometry confirmed clear expression of cell-surface FPN on LSECs, reaching approximately 40% of cells, albeit at lower levels compared to KCs (Fig. 4C). Furthermore, close to 100% of LSECs were positive for HO-1, although the expression levels appeared lower than those in KCs (Fig. 4D). Next, to obtain comprehensive information on LSEC adaptation to Hb clearance, we performed label-free proteomic profiling of FACS-sorted LSECs in comparison with iron recycling macrophages (RPMs and KCs) and splenic and heart ECs. This analysis identified 417 protein groups that were significantly more abundant in LSECs than in splenic and cardiac ECs (Fig. 4E; Dataset EV1). In addition to well-established LSEC markers (STAB1, STAB2, CD206), intriguingly, these included L and H ferritin, HO-1 and BLVRB, heme binding protein HEBP1, and enzymes involved in the antioxidant response (e.g., NQO1, GCLC, or PRDX1). Interestingly, the hits also included several known macropinocytosis effectors, such as Rabankirin-5, ATP6V0A1, RAB5 or septin 2 (Dolat and Spiliotis, 2016; Maxson et al, 2021; Ramirez et al, 2019; Schnatwinkel et al, 2004), and several putative players identified by previous RNAi screens (Ramirez et al, 2019) (Dataset EV1). This proteomic signature of LSEC, indicating their steady-state involvement in Hb uptake and iron recycling from heme, was consistent with their iron/heme indices. As measured by ICP-OES, FACS-sorted LSECs showed total iron levels intermediate between KCs and heart or spleen ECs. Moreover, LSECs exhibited higher labile iron levels compared to other ECs, as determined with Fe$^{2+}$-specific fluorescent probe and flow cytometry (Fig. 4F,G). Likewise, magnetically-sorted LSECs showed slightly lower, though not statistically significant, heme content compared with KCs and splenic RPMs (Fig. 4H). The purity of LSECs and KCs obtained by magnetic separation is shown in Appendix Fig. S3. FPN was robustly induced in LSECs under conditions of systemic iron deficiency, mimicking the response observed in KCs (Fig. 4I), suggesting a tight control of LSECs' FPN by circulating hepcidin. Confirming the iron export function of LSEC FPN, we detected a significant increase in labile iron in LSECs in response to mini-hepcidin PR73 injection, a response that was comparable to that found in KCs (Fig. 4J). Collectively, these data indicate that LSECs are equipped with protein machinery that drives iron recycling from heme, suggesting their role in steady-state iron turnover.

## LSECs and KCs support iron recycling by removing hemolysis products from the spleen

Next, we sought to understand how LSECs may be involved in physiological iron recycling. The hemolysis-driven iron recycling model proposed by Klei et al implied but did not formally demonstrate that RPMs likely sequester spleen-derived Hb via

highly expressed CD163 (Klei et al, 2020). However, CD163 knock-out mice did not show major differences in systemic and splenic iron parameters (Fig. EV3). Inspired by the high capacity of LSECs for Hb scavenging, we hypothesized that hepatic cells may play a role in the clearance of splenic hemolytic products delivered via portal circulation. Therefore, we extended previous studies by quantifying the contributions of splenic and hepatic myeloid and endothelial cells to the uptake of intact RBCs, RBCs devoid of cytoplasm (RBC ghosts), and free Hb. To this end, we injected mice with temperature-stressed RBCs, derived from UBI-GFP transgenic mice, a model where GFP is expressed ubiquitously under the human ubiquitin C promoter. GFP-positive stressed RBCs were additionally stained with the membrane label PKH26 (Fig. 5A,B). This approach revealed that RPMs outperformed other splenic cell types, F4/80$^{high}$ CD11b$^{high}$ pre-RPMs, CD11c$^+$ dendritic cells (DCs), monocytes, and splenic ECs in the sequestration of PKH26$^+$GFP$^+$ intact RBCs. An equal percentage of RPMs (approximately 30% of the population) was effective in removing PKH26$^+$GFP$^-$ RBC ghosts, a function that was efficiently supported by splenic pre-RPMs and, to a lesser extent, by DCs, monocytes, or ECs. Interestingly, we found that in the liver, phagocytosis of intact RBCs by KCs was less efficient than the uptake of RBC ghosts (Fig. 5B). Erythrophagocytosis of intact RBCs was negligible in hepatic DCs and LSECs, but they showed some capacity for the sequestration of RBC membranes, albeit lower than that of KCs. Independently, injection of fluorescently labeled RBC ghosts along with free Hb confirmed that KCs and LSECs, rather than RPMs or splenic ECs, respectively, are specialized in the clearance of these two major hemolysis products (Fig. 5C,D). Corroborating these data, we detected an increase in heme levels in the portal vein plasma, and we identified hemoglobin α (HBA) and β (HBB) chains as top proteins upregulated in LSECs at the proteome-wide level after injection of stressed RBCs compared to control PBS-injected mice (Fig. 5E,F; Dataset EV2). Interestingly, among the five proteins that were significantly increased along with HBA and HBB in LSECs following RBC transfusion, we detected two dehydrogenases ALDH1A1 and ALDH1L1 (Fig. 5F), the former with established roles in the detoxification of aldehydes formed by lipid peroxidation (Makia et al, 2011).

Next, we sought to determine whether the ability of LSECs to sequester Hb could be modulated by the altered capacity of splenic RPMs to fully execute erythrophagocytosis. First, we observed that LSECs were more effective at Hb uptake in response to macrophage depletion after clodronate injection (Fig. 5G). Consistently, plasma and kidney heme levels did not significantly differ between control and clodronate-injected mice at the early 1-hour time point following intravenous Hb administration (Fig. 5H,I). Second, we took advantage of the physiological difference in iron parameters between the BALB/c and C57BL/6 J (BL/6 J) mice. We observed a lower erythrophagocytic capacity of RPMs from BALB/c mice

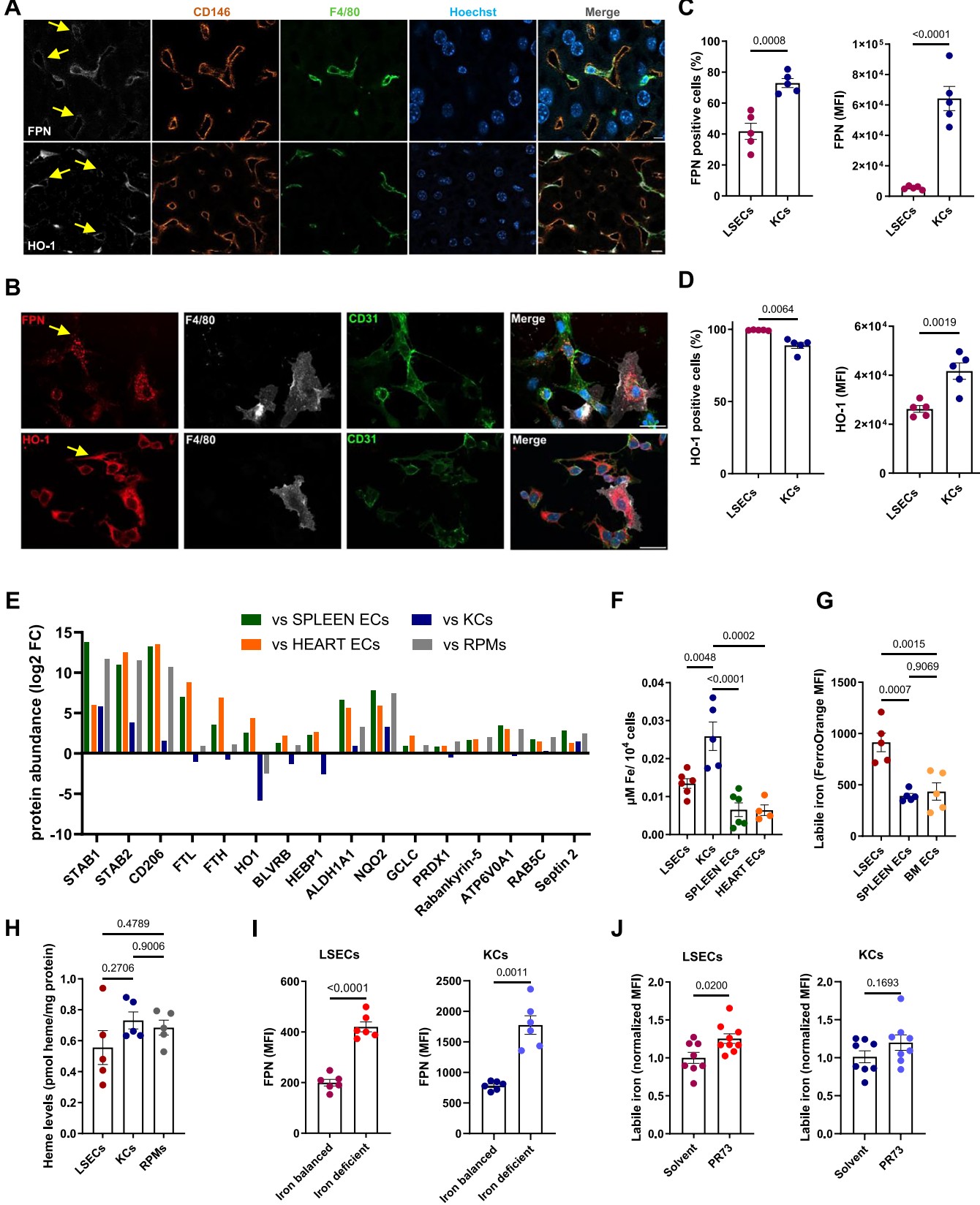

Figure 4.  **LSECs constitutively express proteins critical for iron recycling and macropinocytosis.**

(A) Frozen mouse liver slices were processed and stained for FPN or HO-1 (white) in KCs (F4/80, green) or LSECs (CD146, orange) and imaged using confocal microscopy. Scale bars, 5 µm (upper panel) and 10 µm (bottom panel) (B) FPN and HO-1 presence in F4/80$^+$ (white) KCs and CD31$^+$ endothelial cells (green) imaged by confocal microscopy in murine NPC in vitro cultures. (A, B) Nuclei (blue), scale bars, 20 µm. Arrows indicate ECs positive for FPN or HO-1. (C) Flow cytometry analysis of FPN expression in LSECs and KCs. (D) Flow cytometry analysis of HO-1 expression in LSECs and KCs. (E) Shown are selected proteins that are statistically significantly ($P < 0.01$) more abundant in mouse FACS-sorted LSECs than in spleen and heart endothelial cells (ECs). Log2 fold change (from three independent cell isolates) in protein abundance in LSECs versus the indicated cell type is presented. Data are retrieved from Dataset EV1, where individual label-free quantification values of protein abundance and calculated $P$ values are included. (F) Total iron content in FACS-sorted cells was determined by ICP-OES and normalized to cell numbers. (G) Cytosolic ferrous iron (Fe$^{2+}$) levels in the indicated endothelial cells were measured with a FerroOrange probe and flow cytometry. (H) Heme levels were quantified in magnetically-sorted cells using the Heme Assay Kit and normalized to the protein content. (I) Flow cytometry analysis of FPN expression in LSEC and KCs isolated from mice maintained on balanced or iron-deficient diets. (J) Cytosolic ferrous iron (Fe$^{2+}$) content was measured with a FerroOrange probe and flow cytometry in LSECs and KCs derived from mice injected with mini-hepcidin (PR73). Numerical data are expressed as mean ± SEM, and each data point represents one biological replicate (individual mouse), $n = 5$ (C, D, G, H), 5–6 (F), 6 (I), 8–9 (J). Welch's unpaired $t$ test was used to determine statistical significance in (C, D, I, J); one-way ANOVA with Tukey's Multiple Comparison tests was used in (F, G, H); exact $P$ values are shown on graphs. Source data are available online for this figure.

compared to BL/6 J mice (Fig. 5J), which was likely underlain by higher iron levels and reduced RPM FPN expression (Fig. EV4A,B), two interconnected factors that were previously shown to affect erythrophagocytosis (Slusarczyk et al, 2023). This led to heme accumulation in the extracellular space of the spleen and portal plasma (Figs. 5K and EV4C,D). This phenotype was coupled with elevated levels of labile iron, a higher percentage of HO-1-positive LSECs in the liver, and increased FPN expression in the LSECs of BALB/c mice (Figs. 5L–N and EV4E,F), indicating their enhanced activity in iron recycling from Hb. Taken together, our findings support the physiological role of LSECs in the sequestration of endogenous spleen-derived Hb, thereby establishing a spleen-liver axis for effective iron recycling from senescent RBCs.

## LSECs detoxify hemoglobin upon hemolysis and trigger the iron-sensing BMP6 angiokine

We next investigated the response of LSECs to hemolytic conditions. To this end, we first injected mice with 10 mg of mouse Hb (equivalent to the RBC fraction in approximately 100 µl of blood) in a time-dependent manner. We detected a significant increase in heme iron content in the liver at the early time point, which was normalized by the 24-hour endpoint (Fig. 6A). Consistently, we observed a transient increase in iron levels in the liver, but not in the spleen (Figs. 6B and EV5A). Notably, the kidney also accumulated iron that could not be mobilized and remained elevated throughout the experiment (Fig. EV5B). Hemoglobin clearance in the liver resulted in a strong transcriptional induction of the heme catabolizing enzyme *Hmox1*, a response that could be attributed specifically to LSECs (Fig. 6C,D). At the protein level, we found that 6 h after intravenous Hb injection, HO-1, which under control conditions was predominantly expressed by KCs, appeared mildly reduced in these cells, while levels in LSECs became more comparable, indicating a greater contribution of LSECs to liver HO-1 expression under Hb exposure (Fig. 6E). Notably, although HO-1 was detected in structures resembling sinusoids, the CD146 signal decreased following Hb delivery, an observation that may warrant further investigation. Consistently, we observed that 20 min after high-dose intravenous Hb administration, LSECs, but not KCs or RPMs, showed a significant increase in total iron content, as determined by ICP-OES in magnetically sorted populations (Fig. 6F). The rapid induction of *Hmox1* in LSECs upon intravenous Hb delivery was phenocopied

by injecting the mice with the hemolytic agent phenylhydrazine (PHZ), exceeding the transcriptional response of KCs, splenic cells, and the aorta (Fig. 6G,H). PHZ also caused significant, albeit mild, iron accumulation in the liver (Fig. 6I), but not in other organs (Fig. EV5C,D).

Hemoglobin challenge rapidly increased the hepatic expression of the iron-sensing gene *Bmp6*, attributable to FACS-sorted LSECs, accompanied by transient upregulation of the BMP target gene hepcidin (*Hamp*) in the liver (Fig. 7A–C). As expected, induction of the BMP6–hepcidin axis caused serum hypoferremia at the 15-h time-point, which normalized after 24 h (Fig. 7D), reflecting the changes in liver iron content (Fig. 6B). PHZ-induced hemolysis likewise led to the activation of *Bmp6* transcription in sorted LSECs (Fig. 7E). Finally, we aimed to determine whether LSECs induce a specific response when exposed to Hb compared with non-heme iron. To this end, we performed RNA sequencing to assess global gene expression signatures upon injection of 10 mg of Hb and compared these to a previously reported response to a quantitatively matched dose of iron citrate (Zurawska et al, 2024). Both stimuli induced *Bmp6* at very comparable levels and triggered an ETS1-controlled transcriptional program, as previously reported for iron citrate response (Zurawska et al, 2024) (Appendix Fig. S4). However, the two iron sources elicited different responses of LSECs at the transcriptome-wide level (Fig. 7F). While only 64 genes, mainly attributable to the response to oxidative stress and iron, were co-induced by Hb and iron citrate, as many as 372 and 202 genes increased their expression specifically upon Hb and iron citrate injection, respectively (Fig. 7F). These differences were reflected by a distinct functional enrichment within the Hb- and iron-induced transcriptional signature (Fig. 7G). Interestingly, Hb-exposed LSECs specifically induced genes associated with immune and inflammatory responses, such as the phagocyte marker *Cd68*, the chemokines *Ccl24*, *Ccl6*, and *Ccl9*, and importantly, *Il18*, a cytokine implicated in the pathogenesis of hemolytic sickle cell disease (Gupta et al, 2021). Furthermore, Hb ingestion activated gene expression signatures associated with phagocytosis, the PI3K-Akt pathway, growth factor activity, and intracellular signaling, enriched categories with several links to actin remodeling and, in some cases, directly to macropinocytosis. Indeed, several genes induced by Hb, such as the receptors *Axl, Csf1r*, and *Met*, the growth factors *Hgf* and *Pdgfa*, and the actin cytoskeleton-interacting factors *Evl* and *Coro1a* are known to play a role in the regulation and/or execution of macropinocytosis (BoseDasgupta et al, 2015; Recouvreux and Commisso, 2017; Visweshwaran et al, 2022; Zdzalik-Bielecka et al, 2021).

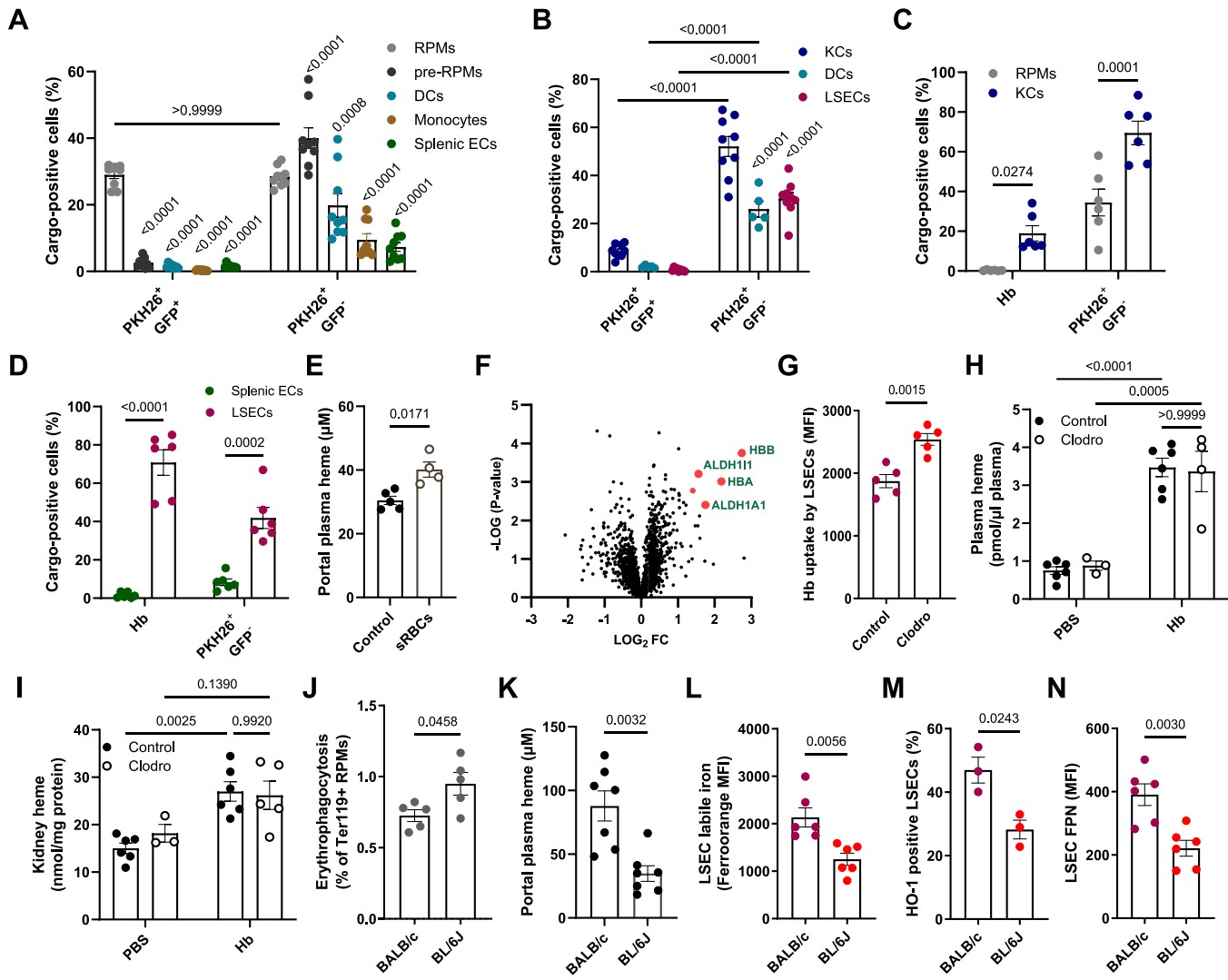

**Figure 5. LSECs and KCs support iron recycling by removing hemolysis products from the spleen.**

(A–D) Mice were administered i.v. with (A, B) stressed GFP⁺ RBCs stained with a membrane marker PKH26 or (C, D) Hb-AF750 and PKH26⁺ RBCs devoid of cytoplasm (RBC ghosts). Splenic and liver cells were isolated, stained, and analyzed by flow cytometry. (A) The percentage of cells positive for markers of intact RBCs (PKH26⁺GFP⁺) and RBC ghosts (PKH26⁺GFP⁻) in splenic RPMs, pre-RPMs, dendritic cells (DCs), monocytes, and endothelial cells (ECs). (B) The percentage of PKH26⁺GFP⁺ and PKH26⁺GFP⁻ cells in liver KCs, dendritic cells (DCs), and LSECs. (C, D) The percentage of cells positive for Hb (AF750) and RBC ghosts. (E) Heme levels from the portal vein plasma of control mice and mice transfused with temperature-stressed RBCs, measured by Heme Assay Kit. (F) Volcano plot illustrating the changes in the proteome of FACS-sorted LSECs 90 min after i.v. transfusion of stressed RBCs. Proteins significantly upregulated are marked in red. Data are included in the Dataset EV2. (G) Hb-AF750 uptake by LSECs derived from control and macrophage-depleted mice (Clodro). (H, I) Heme levels in the plasma (H) and kidney (I) of control and macrophage-depleted mice (Clodro), measured by Heme Assay Kit. (J–N) BALB/c and C57BL/6J (BL/6J) mice phenotype comparison. (J) The capacity of endogenous erythrophagocytosis was assessed by intracellular staining of the erythrocytic marker (Ter119) in RPMs. (K) Heme levels in the portal vein were measured with the Heme Assay Kit. (L) Cytosolic ferrous iron (Fe²⁺) levels in LSECs were measured with a FerroOrange probe and flow cytometry. (M) The percentage of HO-1-positive LSECs was measured with flow cytometry. (N) FPN levels in LSECs were measured by flow cytometry. Data are expressed as mean ± SEM, and each data point represents one biological replicate (individual mouse), $n = 8$–9 (A, B), 6 (C, D, L, N), 4–5 (E), 5 (F, G, J), 3–6 (H, I), 7 (K), 3 (M). Welch's unpaired $t$ test was used to determine statistical significance in (E, G, J–N); while two-way ANOVA with Tukey's Multiple Comparison tests was used in (A–D, H, I); exact $P$ values are shown on graphs, values above bars in (A, B) indicate comparison with RPMs and KCs, respectively, within the same group. Source data are available online for this figure.

Collectively, these findings support the role of LSECs in Hb clearance and demonstrate their high capacity to trigger the iron-sensing *Bmp6* angiokine in response to excessive Hb, along with specific pro-inflammatory markers and factors associated with actin remodeling.

## Discussion

Recent advances have revealed the critical role of specialized hepatic sinusoidal endothelium in maintaining systemic and liver homeostasis. The scavenging activity of LSECs is essential

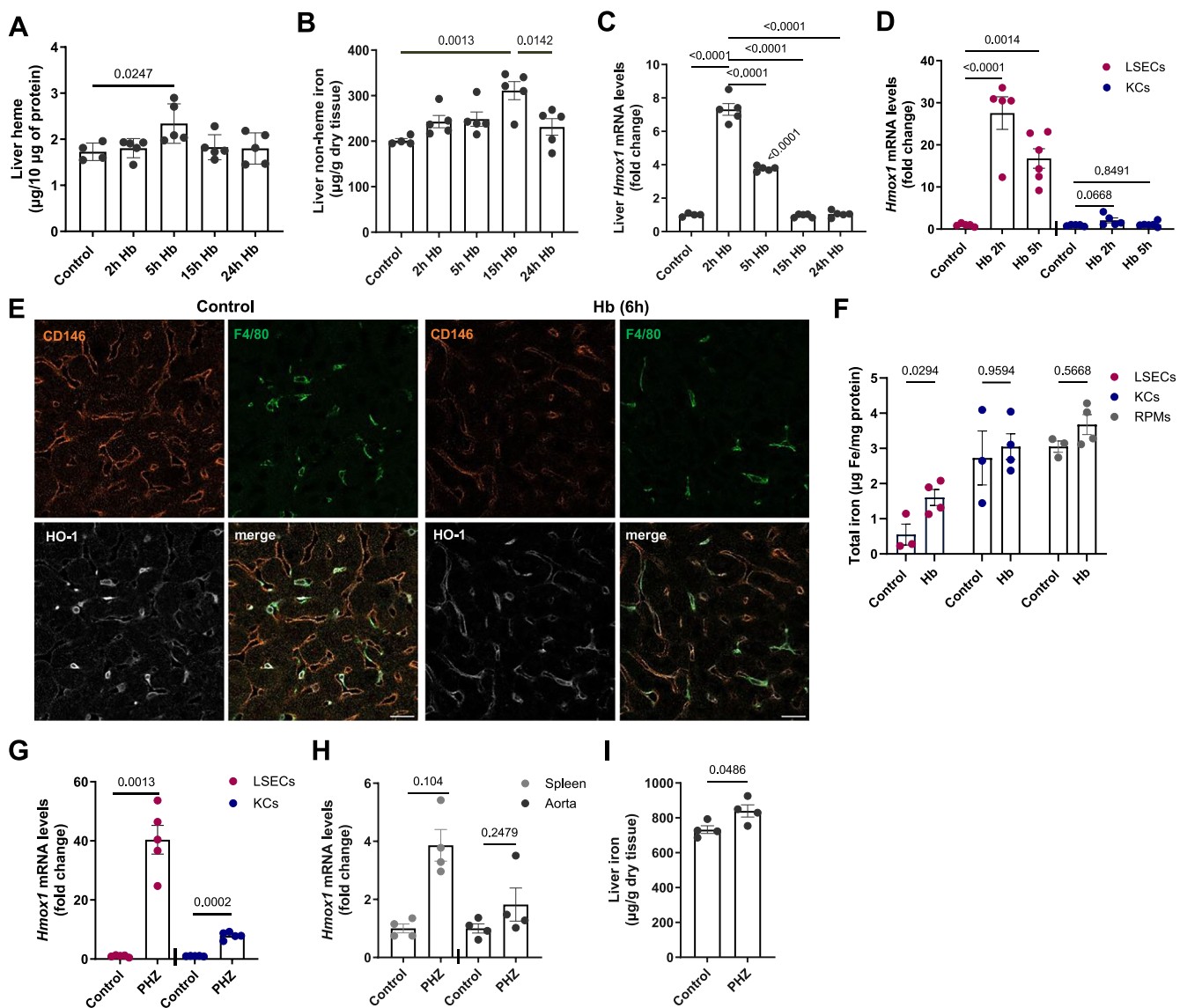

**Figure 6. LSECs detoxify hemoglobin upon hemolysis.**

(A–D) Mice were injected with native hemoglobin (Hb, 10 mg/mouse) and analyzed at the indicated time points. (A) Heme iron levels in the liver were determined with the Heme Assy Kit and normalized to protein content. (B) Non-heme iron content in the liver was measured by bathophenanthroline colorimetric assay. (C, D) *Hmox1* gene expression measurement by RT-PCR in (C) the liver and (D) FACS-sorted KC and LSEC populations. (E, F) Mice were injected i.v. with high-dose Hb (10 mg). (E) Frozen mouse liver slices were processed and stained for HO-1 (white), the LSEC marker CD146 (orange), and F4/80 identifying KCs (green) and imaged using confocal microscopy. Scale bars, 20 μm. (F) Total iron content in magnetically-sorted cells was determined 20 min after injection by ICP-OES and normalized to protein content. (G, H) Hemolysis was induced in mice by i.p. injection of phenylhydrazine (PHZ, 0.125 mg/g) 6 h before analysis. *Hmox1* gene expression in (G) FACS-sorted cell populations of KCs and LSECs, and (H) spleen and aorta. (I) Non-heme iron content in the liver. Data are expressed as mean ± SEM, and each data point represents one biological replicate (individual mouse), n = 4–5 (A–C), 5-6 (D), 3-4 (F), 5 (G), 4 (H, I). One-way ANOVA with Tukey's Multiple Comparison tests was used in (A–C), and separately for KCs and LSECs in (D); two-way ANOVA with Tukey's Multiple Comparison tests was used in (F), and Welch's unpaired *t* test was used to determine statistical significance in (G–I), separately for each cell type/organ; exact *P* values are shown on graphs, a value above the bar in (C) indicates comparison with control mice. Source data are available online for this figure.

for the hepatic clearance of biological macromolecules, such as denatured collagen, glycans/glycation end products, modified low-density lipoproteins, small immune complexes, lipopolysaccharides, and viruses (Ganesan et al, 2012; Koch et al, 2021; Malovic et al, 2007; Schledzewski et al, 2011; Shetty et al, 2018). In addition, LSECs regulate the vascular tone in the liver, balance the tolerogenic immune milieu with the onset of immune responses, ensure physiological zonation of hepatic immune cells (Gola et al, 2021; Shetty et al, 2018), and control iron balance by producing the angiokines BMP2 and BMP6 (Canali et al, 2017; Koch et al, 2017). Our study identified the novel homeostatic role of LSECs in the clearance of free Hb, an activity in which their scavenging and iron-regulatory functions converge.

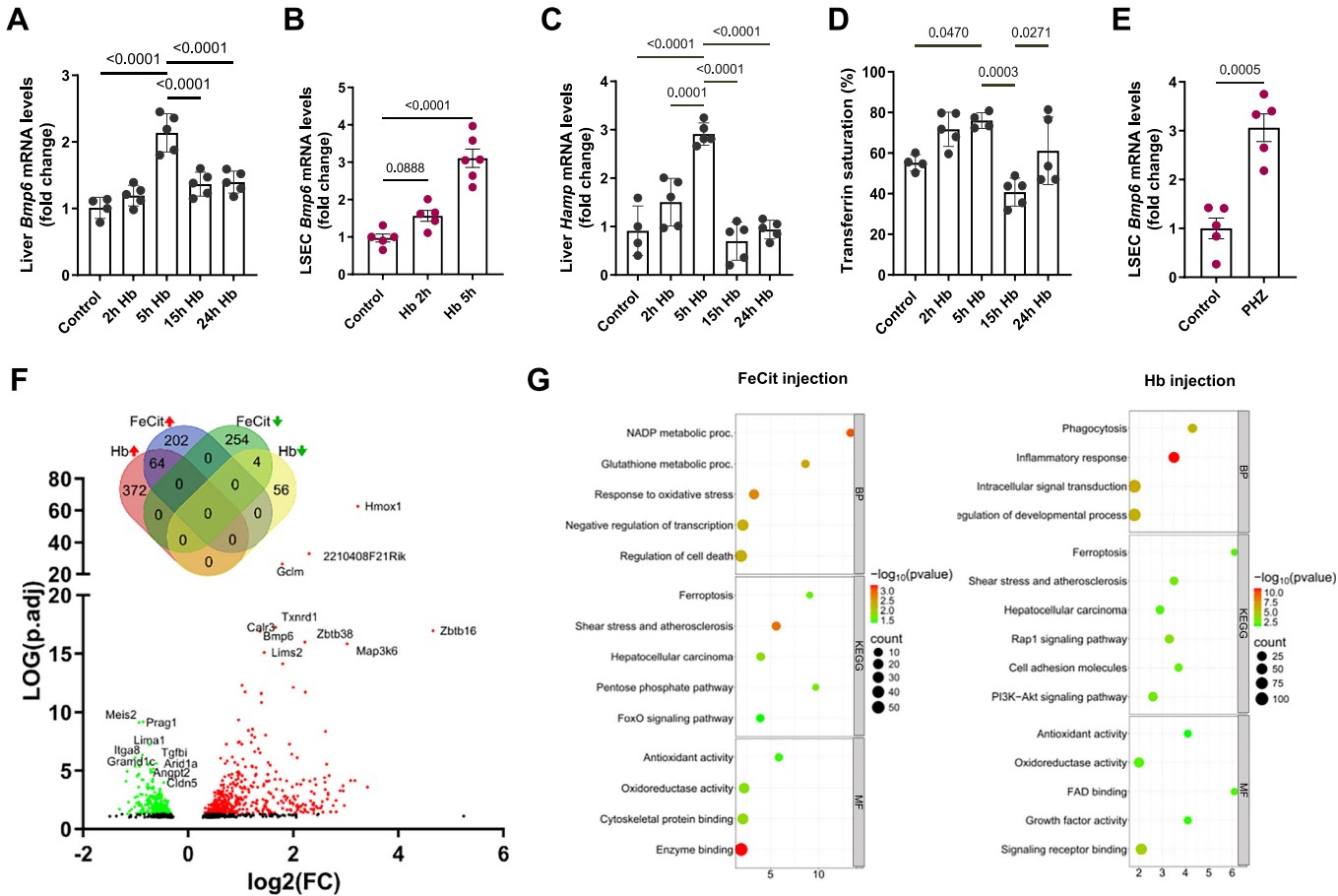

**Figure 7. LSECs trigger the iron-sensing BMP6 angiokine upon hemolysis.**

(A–D) Mice were injected with native hemoglobin (Hb, 10 mg/mouse) and analyzed at the indicated time points. (A, B) *Bmp6* and (C) *Hamp* gene expression measured by RT-qPCR in whole livers or FACS-sorted LSECs, as indicated. (D) Plasma iron levels were determined by transferrin saturation measurements. (E) *Bmp6* expression in FACS-sorted LSECs in mice injected with (PHZ, 0.125 mg/g, 6 h). (F, G) Mice were injected with Hb (10 mg) or Ferric citrate (FeCit, 150 μg) for 5 h. (F) A volcano plot of differentially regulated genes identified by the RNA-seq transcriptomic analysis in FACS-sorted LSECs. The green color indicates negatively, and the red color positively-regulated genes. A Venn diagram shows genes regulated by Hb and FeCit. (G) functional enrichment among genes induced by Hb or FeCit. The *x* axis represents the fold of enrichment. The *y* axis represents GO terms of biological processes (BP), Kyoto Encyclopedia of Genes and Genomes terms (KEGG), and GO terms of molecular function (MF). The size of the dot represents the number of genes under a specific term. The color of the dots represents the adjusted *P*-value. Data are expressed as mean ± SEM, and each data point represents one biological replicate (individual mouse), *n* = 4–5 (A, C, D), 5–6 (B), 5 (E), 4 (F). One-way ANOVA with Tukey's Multiple Comparison tests was used in (A–D), while Welch's unpaired *t* test was used to determine statistical significance in (E); exact *P* values are shown on graphs. Source data are available online for this figure.

The clearance functions of LSECs have been attributed to their extraordinary endocytic activity and surface expression of scavenger receptors (Koch et al, 2021). We discovered that LSECs have a remarkable and constitutive capacity for macropinocytosis, similar to that of macrophages, which use this pathway for antigen presentation, and in contrast to the inducible macropinocytosis of cancer cells, which is critical for nutrient acquisition (Canton, 2018; Commisso et al, 2013). We showed that this capability distinguishes LSECs from ECs of other organs, thus significantly expanding the scarce knowledge on the macropinocytic activity of ECs (Lin et al, 2020) and assigning physiological significance to early observations that LSECs can engage macropinocytosis for antigen capture (Connolly et al, 2010). We propose that the macropinocytosis of LSECs likely contributes to their scavenging function. While this possibility merits future studies, macropinocytosis has recently been identified as a major driver of LDL uptake by macrophages,

leading to foam cell formation in atherosclerosis (Lin et al, 2022). Hence, our findings aid in shifting the paradigm from the sole role of scavenger receptor-mediated endocytosis in the clearance of blood-borne macromolecules, emphasizing the importance of macropinocytic uptake. Finally, our data also demonstrated that the macropinocytic activity of KCs supports their uptake of both free Hb and Hb complexed with Hp, broadening the significance of this route in Hb clearance.

Our study employed in vivo approaches to verify the role of macrophages in Hb uptake and revealed that LSECs qualitatively and quantitatively outcompete CD163-expressing KCs and RPMs in this task. These provocative findings may be explained by the fact that the role of CD163 in Hb uptake has been mainly investigated by its ectopic expression in non-macrophage cells and by using polyclonal anti-CD163 blocking antibodies in cultured macrophages (Kristiansen et al, 2001; Schaer et al, 2006a; Schaer

et al, 2006b). It is noteworthy that CD163-expressing macrophages play important anti-inflammatory roles, such as in the tumor microenvironment or arthritis, but these tissue-specific roles do not appear to be related to Hb clearance (Etzerodt et al, 2020; Etzerodt et al, 2019; Svendsen et al, 2020).

Most importantly, our study defined and quantified the individual contributions of splenic and hepatic cell types to physiological iron recycling. Consistent with previous studies (Ma et al, 2021; Youssef et al, 2018), we provide evidence that RPMs are efficient at phagocytosis of intact stressed RBCs, whereas the liver is the site of scavenging steady-state hemolytic products. KCs were efficient in the uptake of RBC ghosts and outperformed RPMs in Hb uptake. LSECs, owing to their anatomical location, exceptional macropinocytic activity, and expression of iron-recycling proteins, have emerged as a novel cell type involved in the maintenance of iron homeostasis, specialized for the removal of spleen-derived Hb. Consistent with this newly proposed role of LSECs, endothelial-specific FPN knock-out mice have been reported to exhibit marked iron deficiency, anemia, and iron loading in liver NPCs (Zhang et al, 2014).

Importantly, our study has several limitations. Although we propose that macropinocytosis may contribute to the scavenging function of LSECs, this possibility requires further validation. In addition to β-catenin, additional signaling pathways and effector proteins are likely to support the high macropinocytic activity of LSECs. Defining these regulatory mechanisms would allow for their genetic or pharmacological blockage in mice, thereby revealing the consequences of such intervention for LSEC scavenging functions, elimination of body waste products, and iron homeostasis. Furthermore, our study cannot exclude the long-term contribution of iron-recycling macrophages in Hb uptake under hemolytic conditions, nor can we rule out that other tissues or endothelial cell populations may be affected by prolonged or pathological exposure to free Hb and heme. The extent to which LSEC-mediated Hb uptake protects other organs in hemolytic disorders, such as sickle cell disease or thalassemia, remains unresolved. Future studies employing cell-type-specific genetic deletions of proteins involved in iron recycling will be essential to precisely define the relative contributions of RPMs, KCs, and LSECs to systemic iron homeostasis. Finally, the contribution of LSECs to Hb clearance is expected to depend on the degree of splenic hemolysis, which may be modulated by sex, age, pathological conditions, and disease states that affect RPM fitness, factors that remain to be fully elucidated.

# Methods

### Reagents and tools table

| Reagent/resource | Reference or source | Identifier or catalog number |
| --- | --- | --- |
| **Experimental models** | | |
| BALB/c (*M. musculus*) | Experimental Medicine Centre of the Medical University of Bialystok or Mossakowski Medical Research Institute | N / A |
| C57BL/6J (*M. musculus*) | Experimental Medicine Centre of the Medical University of Bialystok | N / A |
| C57BL/6-Tg(UBC-GFP) 30Scha/J (*M. musculus*) | Faculty of Biology, University of Warsaw | IMSR_JAX:004353 |

| Reagent/resource | Reference or source | Identifier or catalog number |
| --- | --- | --- |
| *Cd163−/−* (*M. musculus*) | Department of Biomedicine, Aarhus University | N / A |
| *Primary mouse hepatic cells* | This study | N / A |
| *Primary human hepatic cells* | This study | N / A |
| **Antibodies** | | |
| Anti-mouse CD45 Pe-Cy7 | BioLegend | #103114 |
| Anti-mouse CD45 Pacific Blue | BioLegend | #103126 |
| Anti-mouse CD45 APC-Cy7 | BioLegend | #103116 |
| Anti-mouse CD45 PerCP | BioLegend | #103129 |
| Anti-mouse F4/80 APC-Cy7 | BioLegend | #123118 |
| Anti-mouse CD11b PerCP | BioLegend | #1012300 |
| Anti-mouse CD11b BrilliantViolet 605 | BioLegend | #101237 |
| Anti-mouse CD146 PE | BioLegend | #134704 |
| Rabbit anti-mouse STAB2 AF647 | St John's Laboratory | #STJ192359 |
| Anti-mouse VCAM PE | BioLegend | #105713 |
| Anti-mouse CD31 FITC | BioLegend | #102506 |
| Anti-mouse CD31 APC | BioLegend | #102509 |
| Anti-mouse CD31 PE-Cy7 | BioLegend | #102524 |
| Anti-CD45R/B220 Pacific Blue | BioLegend | #103227 |
| Anti-CD3 Pacific Blue | BioLegend | #100213 |
| Anti-Gr1 Pacific Blue | BioLegend | #108430 |
| Anti-mouse F4/80 PE | BioLegend | #123110 |
| Anti-mouse HO1 | Enzo | #ADI-OSA-150-F |
| Anti-mouse BLVRB | Proteintech | #17727-1-AP |
| Anti-mouse EEA1 | Enzo Life Sciences | #ALX-210-239 |
| Anti-mouse FPN | Amgen | clone 1C7 |
| Anti-mouse Ter-119 | BioLegend | #116201 |
| Anti-mouse Ter-119 488 | BioLegend | #116215 |
| Anti-mouse Ter-119 Pacific Blue | BioLegend | #116231 |
| Donkey anti-rabbit AlexaFluor 647 | Thermo Fisher Scientific | #A-31571 |
| donkey anti-rabbit AlexaFluor 488 | Thermo Fisher Scientific | #A-21206 |
| Anti-human CD36 PE | BioLegend | #336206 |
| Anti-human CD32B PE | BioLegend | #398404 |
| Anti-human CD36 BUV421 | BioLegend | # 336230 |

| Reagent/resource | Reference or source | Identifier or catalog number |
|---|---|---|
| Donkey anti-rabbit IgG 488 | Thermo Fisher Scientific | #A-21206 |
| **Oligonucleotides and other sequence-based reagents** | **Primer Fw (5'- > 3')** | **Primer Rev (5'- > 3')** |
| *Rpl19* | AGGCATATGGG CATAGGGAAGAG | TTGACCTTCAGG TACAGGCTGTG |
| *Bmp6* | ATGGCAGGAC TGGATCATTGC | CCATCACAGT AGTTGGCAGCG |
| *Hmox-1* | AGGCTAAGAC CGCCTTCCT | TGTGTTCCTCT GTCAGCATCA |
| *Hamp* | ATACCAATGCA GAAGAGAAGG | AACAGATACCA CACTGGGAA |
| Random primers | Thermo Fisher Scientific | #48190011 |
| **Chemicals, enzymes and other reagents** | | |
| Clodronate liposomes | LIPOSOMA B.V. | #C-SUV-005 |
| Iron citrate solution (FeCit) | Sigma-Aldrich | #F3388 |
| Mini-hepcidin (PR73) | gift from Elizabeta Nemeth, UCLA | N/A |
| Phenylhydrazine (PHZ) | Sigma-Aldrich | #P26252 |
| PKH26 | Sigma-Aldrich | #MINI26-1KT |
| HBSS | Capricorn | #HBSS-1A |
| Liver Perfusion Medium | Gibco | #17701038 |
| Liver Digest Medium | Gibco | #17703034 |
| Hepatocyte Wash Medium | Gibco | #17704024 |
| Percoll solution | Sigma-Aldrich | #GE17-0891-01 |
| RBCs Lysis Buffer | BioLegend | #420301 |
| Williams E Medium | Sigma-Aldrich | #W1878 |
| FBS | Sigma-Aldrich | # F9665 |
| Penicillin/streptomycin | Thermo Fisher Scientific | # 15140122 |
| Glutamax | Gibco | #35050061 |
| Insulin (human) | Sigma-Aldrich | #I9278 |
| Human Hp | Sigma-Aldrich | #SRP6506 |
| Human hemoglobin | Sigma-Aldrich | #H7379 |
| Collagen I | Corning | # 354236 |
| DPBS | Sigma-Aldrich | # D1408 |
| PBS with Mg²⁺/Ca²⁺ | Gibco | #14040174 |
| EGTA | Sigma-Aldrich | #03777 |
| HBSS | Thermo Fisher Scientific | #14025092 |
| Collagenase P | Roche, Switzerland | #11213873001 |
| Trypan blue | Sigma-Aldrich | #T8154 |
| Chill Protec Plus | Merck | #F2295 |
| AlexaFluor 467 NHS Ester | Thermo Fisher Scientific | #A20006 |
| AlexaFluor 750 NHS Ester | Thermo Fisher Scientific | #A20011 |

| Reagent/resource | Reference or source | Identifier or catalog number |
|---|---|---|
| AlexaFluor 647 Conjugation Kit | Abcam | #ab269823 |
| AlexaFluor 488 Conjugation Kit | Abcam | #ab236553 |
| Dextran Tetramethylrhodamine | Thermo Fisher Scientific | #D1818 |
| Dextran, Texas Red™, 70,000 MW, Lysine Fixable | Thermo Fisher Scientific | #D1864 |
| Dextran, Fluorescein, 70,000 MW, Anionic | Thermo Fisher Scientific | # D1823 |
| Wnt/βcatenin inhibitor PRI-724 | Selleckchem | #S8968 |
| Chlorpromazine (inhibitor of clathrin-mediated endocytosis) | Sigma-Aldrich | #C8138 |
| EIPA (inhibitor of macropinocytosis) | MedChemExpress | #HY-101840 |
| ML141 (Cdc42 inhibitor) | Selleckchem | #S7686 |
| Latrunculin A (actin-depolymerizing agent) | Tocris | #3973 |
| Nystatin (inhibitor of caveolae-mediated endocytosis) | Sigma-Aldrich | # N1638 |
| Human hemoglobin | Sigma-Aldrich | #H7379 |
| Human TruStain FcX™ (Fc Block) | BioLegend | #422302 |
| Anti-F4/80 magnetic beads | Miltenyibiotec | #130-110-443 |
| BSA | Bioshop | #ALB001.250 |
| Mouse TruStain FcX (Fc Block) | BioLegend | #101320 |
| TRIzol™ Reagent | Thermo Fisher Scientific | #15596026 |
| TRIzol™ LS Reagent | Thermo Fisher Scientific | #10296028 |
| Glycogen | Thermo Fisher Scientific | #R0551 |
| RevertAid H Minus Reverse Transcriptase | Thermo Fisher Scientific | #EP0452 |
| SG qPCR Master Mix | Eurex | #E0401-01 |
| RPMI 1640 medium | CAPRICORN | #RPMI-STA |
| Collagenase, Type IV | Sigma-Aldrich | #C5318 |
| DNAse | Sigma-Aldrich | #DN25 |
| Type I collagenase | Thermo Fisher Scientific | # 17100017 |
| Type XI collagenase | Sigma-Aldrich | # C7657 |
| Type I-s hyaluronidase | Sigma-Aldrich | # H3506 |
| Saponin | Sigma-Aldrich | # SAE0073 |
| Gelatin | Sigma-Aldrich | # G7041 |
| Rat serum | Sigma-Aldrich | # R9759 |
| PFA | Thermo Fisher Scientific | #J19943.K2 |
| DMSO | Sigma-Aldrich | #D8418 |
| Sodium bicarbonate | Sigma-Aldrich | #S6014 |

| Reagent/resource | Reference or source | Identifier or catalog number |
|---|---|---|
| Hoechst 33342 | Thermo Fisher Scientific | #H3570 |
| Triton X-100 | Thermo Fisher Scientific | #85111 |
| Phalloidin Atto 390 | Sigma-Aldrich | #50556 |
| ProLong™ Glass Antifade Mountant | Thermo Fisher Scientific | #P36980 |
| Cryomatrix medium | Epredia | #6769006 |
| Citrate-phosphate-dextrose solution with adenine | Sigma-Aldrich | #C4431 |
| Pap Pen Liquid Blocker | Ted Pella | #22311 |
| Lymphosep | Biowest | #L0560-500 |
| DNase I | Sigma-Aldrich | # DN25 |
| Zombie Aqua™ Fixable Viability Kit | BioLegend | #423101 |
| LIVE/DEAD™ Fixable Aqua Cell Stain Kit | Thermo Fisher Scientific | #L34957 |
| LIVE/DEAD™ Fixable Violet Cell Stain Kit | Thermo Fisher Scientific | #L34955 |
| Unsaturated iron-binding kit | Biolabo | #97408 |
| FerroOrange | DojinD | #F374 |
| Heme Assay Kit | Sigma-Aldrich | #MAK316 |
| Collagen I – coated plates | Ibidi | #80841 |
| Amicon Ultra Centrifugal Filter, 10 kDa | Sigma-Aldrich | #UFC9010 |
| **Software** | | |
| Prism software | https://www.graphpad.com | |
| FlowJo Software v10 | https://www.flowjo.com | |
| CytExpert | https://www.mybeckman.pl/ | |
| ZEN 2011 software | https://www.zeiss.com/microscopy/en/products/software/zeiss-zen.html | |
| Adobe Photoshop | https://www.adobe.com/ | |
| ImageJ | https://imagej.net/ij/ | |

## Mice and in vivo procedures

Female BALB/c and C57BL/6J mice (8–10 weeks old) were obtained from the Experimental Medicine Centre of the Medical University of Bialystok or the Mossakowski Medical Research Institute of the Polish Academy of Sciences. Female C57BL/6-Tg(UBC-GFP)30Scha/J (JAX strain #: 004353; UBI-GFP/BL6), which express GFP under control of the human ubiquitin C promoter from a transgenic cassette integrated in chromosome 17, were kindly provided by Aneta Suwińska (Faculty of Biology, University of Warsaw, Poland). Female and male (8–12 weeks old) WT C57BL/6J and Cd163$^{-/-}$ mice were kindly provided by Anders Etzerodt (Department of Biomedicine, Aarhus University, Denmark). For the dietary experiment, C57BL/6J females (4 weeks old) were

obtained from the Experimental Medicine Centre of the Medical University of Bialystok. Mice were fed a standard iron diet of 200 mg/kg (control) or a low iron diet containing <6 mg/kg (iron-deficient) for 5 weeks before analysis. All mice were maintained at the SPF facility under standard conditions (20 °C, humidity 60%, 12-h light/dark cycle). All data on Hb uptake by LSECs were obtained using female mice or primary female LSECs, in line with the previous studies investigating iron recycling mechanisms (Slusarczyk et al; Klei et al). Although we expect these findings to be relevant for both sexes, a detailed comparison of LSEC contribution to iron recycling between sexes would require an independent study. No formal randomization procedure was used; animals/samples were allocated based on the order of collection. No blinding was performed during allocation, conduct, or analysis of the experiments. Experimental design, conduct, and reporting followed the ARRIVE guidelines to ensure transparent reporting of animal research.

### Procedures involving BALB/c mice

Proteins or their conjugates were dissolved in PBS and administered intravenously (i.v.) at doses indicated in the figure legends. For macrophage depletion, mice received an i.v. solution of liposomes containing clodronic acid (LIPOSOMA, #C-SUV-005) (5 ml/kg) or control empty liposomes for 24 h. A sterile aqueous iron citrate (FeCit, 150 µg/mouse) solution (Sigma-Aldrich, #F3388) or sterile citric acid buffer (0.05 M, Sigma-Aldrich, #251275) was normalized to pH 7.0 and administered i.v.; tissues were collected 5 h after injection. Mini-hepcidin (PR73, 50 nmol/mouse) (Stefanova et al, 2017) (kind gift from Elizabeta Nemeth, UCLA, USA) was injected intraperitoneally (i.p.); tissues were harvested 4 h post-injection. To induce hemolysis, a sterile solution of phenylhydrazine (PHZ, Sigma-Aldrich, #P26252) in PBS was administered i.p. at a dose of 0.125 mg/g body weight, and tissues were collected after 6 h.

### Procedures involving C57BL/6J mice

PKH26-stained (Sigma-Aldrich, #PKH26GL-1KT) temperature-stressed UBI-GFP RBCs were resuspended to 50% hematocrit in HBSS (Capricorn, #HBSS-1A) and administered i.v. for 1.5 h in a 100 µl. An equal dose of PKH26-stained RBC ghosts was mixed with fluorescently labeled Hb (10 µg/mouse) and administered i.v. for 1.5 h in a 100 µl. For proteomics analysis of LSECs, wild-type unstained RBCs were resuspended to 50% hematocrit in HBSS and administered i.v. for 1.5 h in a 100 µl.

## Isolation of human non-parenchymal liver cells

Liver tissue samples were obtained from the Department of Hepatobiliary Surgery and Visceral Transplantation at Leipzig University Medical Center. Specimens were collected from macroscopically healthy tissue adjacent to resected segments in patients with primary or secondary liver tumors or benign local liver diseases. During resections, both diseased and surrounding unaffected liver tissue were removed; a portion of this unaffected tissue was used for liver cell isolation. Cells obtained from a total of four patients: a male, aged 73; two females, aged 60; and a female, aged 68, supported the data presented in the figures. Primary human NPCs (hNPCs) were isolated using a two-step collagenase perfusion technique (Damm et al, 2019; Kegel et al, 2016). Briefly,

liver tissue was perfused first with a buffer containing ethylene-diaminetetraacetic acid (EGTA) (Sigma-Aldrich, USA, #03777-10 g), followed by a digestion buffer with collagenase P (Roche, Switzerland, #11213873001). Hepatocytes were separated by washing and centrifuging the cell suspension twice at 50 g in PBS with $Mg^{2+}/Ca^{2+}$ (Gibco, USA, #14040174). The supernatants containing hNPCs were then united and centrifuged at $300\times g$ and subsequently at $650\times g$ (Zimmermann et al, 2021). Cell pellets containing hNPCs were resuspended in PBS with $Mg^{2+}/Ca^{2+}$, and the hNPC fraction was counted and checked for viability using the Trypan blue (Sigma-Aldrich, #T8154) exclusion method. Finally, the hNPC fraction was prepared for shipment by centrifugation, resuspension in Chill Protec Plus (Merck, Germany, #F2295), and transfer into cryovials (Sarstedt, Nümbrecht, Germany, #72.379). After overnight transport at 4 °C, the hNPCs pellet was washed with Williams E Medium (Sigma-Aldrich, #W1878-500ML), containing 10% FBS, and centrifuged for 5 min, 650 g, RT (room temperature). The pellet was resuspended in RBC Lysis Buffer (BioLegend, #420301) and incubated for 5 min. The hNPCs were then washed and centrifuged again at 650 g for 10 min at RT. The cell pellet was resuspended in Williams E Medium (Sigma-Aldrich, #W1878-500ML), containing 10% FBS, 1% Penicillin/Streptomycin, 1% Glutamax (Gibco, #35050061), and human insulin at a concentration of 5 μg/ml (Capricorn, # INS-K) and seeded for experiments on collagen-I-coated plates.

## Primary murine cell culture

Murine liver cells were isolated from female BALB/c mice according to the standard two-step perfusion method with minor modifications. Briefly, mice were euthanized, and livers were perfused in situ (5 ml/min) with Liver Perfusion Medium (Gibco, #17701038) for 3 min through the inferior vena cava after transection of the portal vein. Next, the medium was exchanged for prewarmed (37 °C) Liver Digest Medium (Gibco, #17703034), and perfusion continued for the next 15 min. Digested livers were gently disintegrated in Hepatocyte Wash Medium (Gibco, #17704024) and filtered through a 100-μm cell strainer. The cell suspension was then centrifuged at 50 g for 3 min to separate hepatocytes from the supernatant containing non-parenchymal cells (NPCs). Hepatocytes were further purified by layering them on 1.06 g/mL Percoll solution (Sigma-Aldrich, #GE17-0891-01) and centrifugation at $750\times g$ for 20 min without a break. NPCs were centrifuged two more times at $50\times g$ for 3 min and spun at $650\times g$ for 15 min, 4 °C. The pellet was resuspended in RBCs Lysis Buffer, incubated for 3 min, washed, and centrifuged again for 10 min, $650\times g$, 4 °C. Subsequently, the NPCs were resuspended in Williams E Medium (Sigma-Aldrich, #W1878-500ML), containing 10% FBS, 1% Penicillin/Streptomycin, 1% Glutamax (Gibco, #35050061), and insulin at a concentration of 10 μg/ml (Sigma-Aldrich, #I9278). Cells were seeded on collagen I-coated (Corning, # 354236) plates or glass microscope slides at a density of 30,000 cells/cm². After 3 h, the cells were washed with PBS, and the medium was replaced with Williams E Medium supplemented with 4% FBS, 1% Penicillin/Streptomycin, and 1% Glutamax.

## Cell culture treatments

Murine NPCs primary cells were cultured in Williams E medium supplemented with 4% FBS, 1% Penicillin/Streptomycin, and 1% Glutamax. The day after cell seeding, the cells were stimulated with freshly prepared fluorescently stained hemoglobin (Hb-AF647 or Hb-AF750, 0.5 μg/ml), hemoglobin-haptoglobin complex (Hb-AF750:Hp, 0.5 μg:15 μg/ml), Dextran Tetramethylrhodamine, 70,000 MW, Lysine Fixable (1 mg/ml, ThermoScientific, #D1818), Dextran, Texas Red, 70,000 MW, Neutral (0.25 mg/ml, Thermo-Scientific, # D1830) or Albumin from Bovine Serum (BSA-AF647) (1 μg/ml, ThermoScientific, # A34785) for 1 h. When indicated, the following compounds were used for pre-treatments: Wnt/beta-catenin inhibitor PRI-724 (5 μM, for 24 h, Selleckchem, #S8968); inhibitor of clathrin-mediated endocytosis, chlorpromazine (2 μM, for 1 h, Sigma-Aldrich, #C8138); inhibitor of macropinocytosis, EIPA (25 μM, for 1 h, MedChemExpress, #HY-101840); inhibitor of Cdc42 GTPase, ML141 (10 μM, for 1 h, Selleckchem, #S7686), an actin-depolymerizing agent, latrunculin A (1 μM, for 30 min, Tocris, #3973) and inhibitor of caveolae-mediated endocytosis, nystatin (240 U/ml for 30 min, Sigma-Aldrich, # N1638).

The day after seeding, hNPCs were stimulated with freshly prepared fluorescently stained human hemoglobin (Sigma, #H7379; Hb-AF647, 0.5 μg/ml for immunofluorescence imaging, 5 μg/ml for flow cytometry analysis) or Dextran, Texas Red™, 70,000 MW, Lysine Fixable (1 mg/ml, ThermoScientific, # D1864) for 1 h. When indicated, the following compounds were used for pre-treatments: chlorpromazine (2 μM, for 1 h, Sigma-Aldrich, #C8138); EIPA (25 μM, for 1 h, MedChemExpress, #HY-101840), and latrunculin A (1 μM, for 30 min, Tocris, #3973).

## Preparation of single-cell suspension for flow cytometry

For flow cytometry analysis of the liver cells, mice were euthanized, and liver lobes were dissected. Next, liver lobes were perfused with PBS until blood removal, minced with scissors, and enzymatically digested in RPMI 1640 medium (CAPRICORN, #RPMI-STA) containing type IV collagenase (1 mg/ml, Sigma-Aldrich, #C5318-5G) and DNAse (20 mg/ml, Sigma-Aldrich, #DN25-1G) for 40 min at 37 °C with shaking. Next, the cell suspension was passed through a 100 μm cell strainer, washed with PBS, and centrifuged 2× at $50\times g$, 3 min, RT to remove hepatocyte pellets. Next, the supernatant was centrifuged at $650\times g$, 15 min, and 4 °C. Cells were suspended in RBCs Lysis Buffer, incubated for 5 min, RT, then washed with PBS and centrifuged at $650\times g$, 15 min, 4 °C. For the splenocytes, the spleen was minced with scissors and subjected to digestion under similar conditions as the liver, but with a shorter incubation time of 20 min at 37 °C. Next, the organ pieces were passed through a 100 μm cell separation strainer, washed with PBS, and centrifuged at $500\times g$, 5 min, and 4 °C. Cells were suspended in RBCs Lysis Buffer, incubated for 7 min, RT, then washed with PBS and centrifuged at $500\times g$, 5 min, 4 °C. For the analysis of femurs and tibias, the epiphysis of the dissected bones was cut off, and the bones were placed vertically in a specially cut tip for an automatic pipette placed in an Eppendorf tube, so that the bone did not touch the bottom. The tubes were centrifuged at $1000\times g$, 1 min, RT. The cell pellet was suspended in RBCs Lysis Buffer and incubated for 5 min at RT, washed with PBS, and centrifuged at $500\times g$, 5 min, 4 °C. The cell pellet was resuspended in the same enzymatic mixture used for liver and spleen digestion and incubated for 15 min at 37 °C with shaking. The cells were then washed with PBS and centrifuged at $500\times g$, 5 min, and 4 °C. The aorta was dissected and minced with scissors. Next tissue was enzymatically digested with a mix of enzymes: type I collagenase, type XI collagenase, type

I-s hyaluronidase, and DNase I according to the protocol published previously (Gjurich et al, 2015). The organ pieces were then filtered through a 100 μm cell separation strainer, rinsed with PBS, and centrifuged at 500× *g*, 5 min, and 4 °C.

## Isolation of liver non-parenchymal cells, spleen, and heart endothelial cells for FACS sorting

For FACS-sorting of murine LSECs and KCs, livers were perfused as described above with an additional step of purification. Briefly, NPCs depleted of RBCs were mixed with 1.06 g/mL Percoll (Sigma-Aldrich, #GE17-0891-01) solution (1:1) and centrifuged at 800× *g*, 30 min at RT (without brake). Cell pellets were resuspended in PBS and stained with LIVE/DEAD™ Fixable Violet Cell Stain Kit (ThermoScientific, #L34955) according to the manufacturer's instructions, and centrifuged at 650 g, 5 min, 4 °C. Next, cells were resuspended with FACS buffer (1% BSA in PBS) and incubated with TruStain FcX™ (Fc block; BioLegend, #101320) 1:100 for 5 min. Next, cells were stained with the following antibodies (BioLegend): rat anti-mouse CD45 Pe-Cy7 (#103114), F4/80 APC-Cy7 (#123118), CD11b PerCP (#1012300), CD146 PE (#134704) and unconjugated rabbit-anti-mouse STAB2 (St John's Laboratory, #STJ192359) for 30 min at 4 °C, in 1:100 dilution. Cells were washed and stained with the secondary antibodies donkey-anti-rabbit AlexaFluor 647 (ThermoScientific, #A-31571, dilution 1:200) or donkey-anti-rabbit AlexaFluor 488 (ThermoScientific, #A-21206, dilution 1:200). Next, cells were washed and sorted using BD FACS Aria II sorter (BD Biosciences). For gene expression analysis, cells were sorted directly into TRIzol™ LS Reagent (ThermoScientific, #10296028) at a quantity of 20,000–50,000 cells, with a maximum flow rate of 2000 events per second using an 85 or 100 μm nozzle. The sorting purity was approximately 90% from CD45$^+$ cells. For quantification of cellular total iron levels (ICP-OES), cells were sorted into PBS and spun for 5 min at 1000× *g* (at a quantity of 40,000–200,000 cells).

For FACS sorting of spleen and heart endothelial cells, mice were euthanized and perfused via heart with ice cold PBS containing heparin (1:100). Next, organs were minced with scissors and enzymatically digested in RPMI 1640 medium containing type IV collagenase (1 mg/ml) and DNAse (20 mg/ml) for 20 (spleen) or 40 (heart) min at 37 °C with shaking. Next, the cell suspension was passed through a 100 μm cell strainer, washed with PBS, and resuspended in RBCs Lysis Buffer. After 5 min of incubation, the cells were washed with PBS and centrifuged at 650× *g*, 5 min at 4 °C. The Fc block was added at a 1:100 dilution in FACS buffer and incubated at 4 °C for 10 min. The cells were then stained with the following antibody panel for 30 min at 4 °C in the dark in 1:100 dilution: CD45 (Biotin, BioLegend, #103104), together with a trace "spike-in" of CD45 (PE-Cy7, BioLegend, #103114) to monitor depletion efficiency by flow cytometry. CD3 (BV421, BioLegend, #100227), Gr-1 (BV421, BioLegend, #108433), and B220 (BV421, BioLegend, #103245), F4/80 (PE, BioLegend, #123110), and CD11b (FITC, BioLegend, #101206) and CD31 (APC, BioLegend, #102410) to identify endothelial cells. Following incubation, the cells were washed with cold PBS containing 0.5% BSA (sorting buffer) and centrifuged at 650× *g* for 5 min. The pellet was resuspended in 200 μL of sorting buffer containing 50 μL of MojoSort Streptavidin Nanobeads (BioLegend, #480016) and incubated at 4 °C in the dark for 20 min. After incubation, 2.5 mL of sorting buffer was added,

and the tube was placed on an EasyEights EasySep Magnet (STEMCELL, #18103) for 10 min. The supernatant, corresponding to the CD45$^-$ fraction, was transferred to a fresh 5 mL tube and centrifuged at 650× *g* for 5 min, and resuspended in sorting buffer. For quantification of cellular total iron levels (ICP-OES), cells were sorted into PBS and spun for 5 min at 1000× *g*. Gating strategy details are reported in the Appendix.

## Flow cytometry

Murine cell pellets were resuspended in PBS and stained with LIVE/DEAD™ Fixable Violet/Aqua Cell Stain Kit (ThermoScientific, #L34955, # L34957) or Zombie Aqua™ Fixable Viability Kit (BioLegend, #423101) according to the manufacturer's instructions, and centrifuged at 650× *g*, 5 min, 4 °C. Next, cells were resuspended with FACS buffer (1% BSA in PBS) and incubated with rat serum (5%) and Fc block 1:100 for 5 min. In the next step, liver cells were stained with the following rat anti-mouse antibodies (BioLegend): anti-CD45 Pe-Cy7 (#103114), anti-F4/80 APC-Cy7 (#123118), anti-CD11b PerCP (#1012300), anti-CD146 PE (#134704) and unconjugated rabbit-anti-mouse STAB2 (St John's Laboratory, #STJ192359) for 30 min at 4 °C, in 1:100 dilution. Secondary antibody staining was performed with donkey-anti-rabbit IgG AlexaFluor 647 (ThermoScientific, #A-31571, dilution 1:200) or donkey-anti-rabbit IgG AlexaFluor 488 (ThermoScientific, #A-21206, dilution 1:200). After discrimination of dead cells and doublets, LSEC cells were gated as a population moderately expressing CD45, lacking macrophage-specific proteins F4/80 and CD11b, and expressing STAB2 and CD146 receptors. KCs were gated as a population with high expression of CD45 and F4/80 and moderate expression of CD11b.

For analysis of the splenic cell populations, the following surface rat anti-mouse BioLegend antibodies were used: anti-CD45 PerCP (#103129), anti-F4/80 PE (#123109), anti-CD11b BrilliantViolet 605 (#101237), anti-CD45R/B220 Pacific Blue (#103227), anti-CD3 Pacific Blue (#100213), anti-Gr1 Pacific Blue (#108430), anti-Ter119 Pacific Blue (#116231) and anti-CD31 PE-Cy7 (#102524). Splenic RPMs were gated as a population negative for CD3, Gr1, B220 and Ter119, with high expression of CD45, F4/80, and moderate expression of CD11b. Splenic endothelial cells were gated as a population negative for CD45, expressing CD31. For analysis bone marrow macrophages and endothelial cells (ECs) the following surface rat anti-mouse BioLegend antibodies were used: anti-CD45 PerCP (#103129), anti-CD11b BrilliantViolet 605 (#101237), anti-VCAM PE (#105713), anti-CD45R/B220 Pacific Blue (#103227), anti-CD3 Pacific Blue (#100213), anti-Gr1 Pacific Blue (#108430), anti-Ter119 Pacific Blue (#116231) and anti-CD31 PE-Cy7 (#102524). Macrophages were gated as a population negative for CD3, Gr1, B220, and Ter119, with high expression of CD45, VCAM, and moderate expression of CD11b. ECs were gated as a population negative for CD45, expressing CD31. Immediately before the analysis, the cell suspension was transferred to tubes with a sieve with a pore diameter of 35 μm.

The content of intracellular ferrous iron (LIP, Fe$^{2+}$) was measured using FerroOrange (DojinD, #F374). Briefly, surface-stained cells were incubated with 1 μM FerroOrange in HBSS for 30 min at 37 °C and analyzed directly by flow cytometry without further washing. Detection of FPN was performed with a non-commercial antibody that recognizes the extracellular loop of

mouse FPN [rat monoclonal antibody, Amgen, clone 1C7; directly conjugated using AlexaFluor 488 Labeling Kit (Abcam, ab236553)] (Slusarczyk et al, 2023). For intracellular HO-1 staining, surface-stained cells were fixed with 4% PFA and permeabilized with 0.5% Triton-X in PBS. Next, cells were stained for 30 min at 4 °C with primary anti-HO-1 (ENZO, #ADI-OSA-150-F) conjugated with AlexaFluor 488. Conjugation was performed with the Conjugation Kit (Abcam, #ab236553) according to the manufacturer's protocol. Erythrophagocytosis capacity in RPMs was determined by intracellular staining of the erythrocytic marker using anti-Ter-119 (BioLegend, #116201 and #116215) as described previously (Slusarczyk et al, 2023). The panel of antibodies used for FPN and HO-1 staining consisted of anti-CD45 PE-Cy7, anti-F4/80 anti-APC-Cy7, anti-CD11b PerCP, anti-CD146 PE, and anti-STAB2-AlexaFluor 647 (Gbiosciences, #ITN 2255-647). Due to the wide emission spectrum, the panel employed for experiments involving FerroOrange was modified by omitting anti-CD11b and anti-CD146 markers: for liver NPCs, it included anti-CD45 Pacific Blue (#103126), anti-F4/80 APC-Cy7 (#123118), anti-STAB2-AlexaFluor 647 (Gbiosciences, #ITN 2255-647) or anti-STAB2 (St John's Laboratory, # STJ192359), while for splenic and bone marrow ECs, it included anti-CD45 APC-Cy7 (#103116), anti-CD31 APC (#102509), anti-CD45R/B220 Pacific Blue (#103227), anti-CD3 Pacific Blue (#100213), anti-Gr1 Pacific Blue (#108430), and anti-Ter119 Pacific Blue (#116231).

After hemoglobin treatment, human hNPCs were washed with FACS buffer and centrifuged at 650× $g$, 5 min, 4 °C. hNPCs pellets were stained with LIVE/DEAD™ Fixable Aqua Cell Stain Kit (ThermoScientific, # L34957) according to the manufacturer's instructions, and centrifuged at 650× $g$, 5 min, 4 °C. Next, cells were resuspended with FACS buffer and incubated with Human TruStain FcX™ (BioLegend, # 422302) 1:100 for 5 min. In the next step, human liver cells were stained with mouse anti-human CD36 BUV421 (Biolegend, # 336230) 1:100 and CD32B PE (Biolegend, #398404) 1:100 antibodies for 1 h at 4 °C. After staining, cells were washed with FACS buffer and centrifuged at 650× $g$, 5 min, 4 °C. Pellets were fixed with 4% PFA for 15 min at RT, washed with FACS buffer, and centrifuged at 750× $g$, 5 min, 4 °C.

Analyses were performed using BD LSRFortessa X-20 (BD Biosciences), BD Canto II (BD Biosciences), BD Aria II sorter (BD Biosciences) or CytoFLEX (Beckman Coulter) flow cytometers and were analyzed with FlowJo or CytExpert, respectively. The geometric mean fluorescence intensities (MFI) corresponding to the probes/target protein levels were determined. For quantifications, the MFI of the adequate fluorescence minus one (FMO) control was subtracted from the sample's MFI, and data were further normalized. All gating strategy details are reported in the Appendix.

## Magnetic sorting of liver sinusoidal endothelial cells, Kupffer cells and red pulp macrophages

NPC pellets obtained after liver perfusion were resuspended in RBCs Lysis Buffer, incubated for 4 min, washed with sorting buffer (0,5% BSA/PBS), and centrifuged for 10 min, 650× $g$, 4 °C. NPCs were further purified using the Percoll gradient centrifugation protocol. Briefly, NPCs pellet was resuspended in 3 ml of sorting buffer, layered on the top of a two-layer Percoll gradient (3 ml of 50% Percoll on the bottom, 3 ml of 25% Percoll on the top) and

centrifuged at 800 g without brake, 20 min, RT. NPCs were mainly localized and collected from the interphase between 25% and 50% Percoll, washed with sorting buffer, and spun at 700× $g$ for 10 min. The pellet was resuspended in 180 μl of sorting buffer containing Fc block 1:100 and incubated on ice for 5 min. Next, cells were mixed with 30 μl of anti-F4/80 MicroBeads UltraPure magnetic beads (Miltenyi Biotec, #130-110-443) and incubated for 15 min on ice with regular shaking. After that, cells were washed and spun for 5 min, 400× $g$, 4 °C. The cell pellet was resuspended in sorting buffer and passed through a magnetized LS Separation Column (Miltenyi Biotec, # 130-042-401). F4/80 positive cells (Kupffer cells) were eluted from the demagnetized column, washed with sorting buffer, and spun for 5 min, 400× $g$, 4 °C. The flow-through suspension containing F4/80 negative cells was collected and centrifuged for 5 min at 400× $g$, 4 °C. Next, the cell pellet was incubated in the mixture of 180 μl of sorting buffer and 30 μl anti-CD146 (LSEC) MicroBeads (Miltenyi Biotec, # 130-092-007) for 15 min on ice with regular shaking. Cells were washed, spun for 5 min, 400× $g$, 4 °C, and resuspended in sorting buffer. The suspension was loaded on the LS Separation Column to obtain a pure population of LSECs. F4/80$^-$ CD146$^+$ LSECs were eluted from the demagnetized column, washed, and centrifuged for 5 min at 400× $g$, 4 °C. Pellets were washed twice with PBS and incubated with RIPA buffer containing Protease Inhibitor Cocktail (Sigma,#4693124001), followed by centrifugation for 15 min at 12,000× $g$. Supernatant was further collected and saved for heme level or total iron content analysis.

To obtain a magnetically sorted population of red pulp macrophages (RPMs), single-cell suspensions of spleens were prepared. The Fc block was added at a 1:100 dilution in FACS buffer (1% BSA/PBS) and incubated at 4 °C for 10 min. The cells were then labelled with the following antibodies: anti-F4/80 (APC, Biolegend, #123116), anti-Ly-6G/Ly-6C (Gr-1) (Biotin, Biolegend, # 108403), anti-CD3 (Biotin, Biolegend, #100243), and anti-B220 (Biotin, Biolegend, #103204), in the dilution 1:100 for 30 min at 4 °C in the dark. After incubation, cells were washed with cold PBS containing 0.5% BSA (sorting buffer) and centrifuged at 650× $g$ for 5 min. The pellet was resuspended in 200 μL of sorting buffer containing 50 μL of MojoSort Streptavidin Nanobeads (Biolegend, #480016) and incubated at 4 °C in the dark for 20 min. Subsequently, 2.5 mL of sorting buffer was added, and the tube was placed on an EasyEights EasySep Magnet (STEMCELL, #18103) for 10 min. The supernatant was transferred to a fresh 5 mL tube and centrifuged at 650× $g$ for 5 min. The resulting pellet was resuspended in 175 μL of sorting buffer, and 25 μL of MojoSort Mouse anti-APC Nanobeads (Biolegend, #480072) were added. The suspension was gently mixed and incubated for 20 min at 4 °C in the dark. After incubation, 2.5 mL of sorting buffer was added, and the tube was placed on the magnet for 10 min. The bead-bound F4/80$^+$ cells were washed twice with PBS and incubated with RIPA buffer as mentioned above for heme level and total iron content analysis.

## Isolation of murine hemoglobin (Hb) and conjugation of mouse Hb, human Hb and bovine serum albumin (BSA)

To isolate hemoglobin (Hb) from mouse erythrocytes, peripheral blood was collected after euthanasia by cardiac puncture into heparinized tubes. Blood was centrifuged at 500× $g$ for 5 min, RT;

the plasma fraction and the buffy coat were discarded. Erythrocytes were suspended in PBS and centrifuged at 500× *g*, 5 min, and 4 °C, and this washing was repeated four times. Erythrocytes were then centrifuged at 3000× *g*, 5 min, 4 °C. The pellet was resuspended in 20 ml of cold sterile deionized water and incubated overnight at 4 °C. Finally, the solution was centrifuged at 3000× *g*, 5 min, 4 °C, and concentrated using an Amicon® Ultra-15 filter unit with a 10 kDa cutoff (Sigma-Aldrich, # UFC901008D). Hb concentration was calculated from the Beer-Lambert law and heme extinction coefficient (167,000 M − 1·cm−1) at 415 nm, which was measured using a NanoDrop 2000 spectrophotometer (Thermo Fisher Scientific).

AlexaFluor NHS esters AlexaFluor 647 (#A20006) or AlexaFluor 750 (#A20111) (Thermo Fisher Scientific) were dissolved in DMSO and diluted in 0.1 M $NaHCO_3$, pH 8.3. Isolated murine hemoglobin, human hemoglobin (Sigma-Aldrich, # H7379), or BSA were dissolved in 0.1 M $NaHCO_3$, pH 8.3. Protein and ester solutions were mixed in a volume ratio of 1:1 at a molar ratio of 1:2.5 and incubated at 25 °C in the dark for 1 h with shaking. Then the conjugate was washed with 0.1 M $NaHCO_3$, pH 8.3, and concentrated using an Amicon® Ultra-15 filter unit with 10 kDa cut-off at 4000× *g*, 15 min, 4 °C. Hb concentration was calculated from the Beer-Lambert law and heme extinction coefficient (167,000 $M^{-1}·cm^{-1}$) at 415 nm, which was measured using a NanoDrop 2000 spectrophotometer (Thermo Fisher Scientific). BSA concentration was measured using standard methods. The effectiveness of conjugation was confirmed by mass spectrometry analysis. To prepare of Hb-haptoglobin (Hp) complex, Hb-AF750 and human Hp (Sigma-Aldrich, #SRP6506) solutions were mixed in a molar ratio (Hb-AF750:Hp) of 1:6 and incubated for 30 min at 4 °C in the dark. Then the complex was purified from free Hb or Hp by centrifugation using an Amicon® Ultra-0.5 filter unit with 100 kDa cutoff (Sigma-Aldrich, # UFC510024) at 4000× *g*, 3 min, 4 °C.

## RBC preparation and staining

RBC isolation, labeling, and RBC ghost preparations were performed as described previously (Slusarczyk et al, 2023) with some modifications. Briefly, whole blood obtained from the wild-type C57BL/6 J and UBI-GFP/BL6 mice was collected in CPDA-1 solution (Sigma-Aldrich, #C4431). The suspensions were mixed with HBSS, loaded on Lymphosep (Biowest, #L0560-500), and centrifuged at 400× *g* for 15 min at RT. The buffy coat was removed, and the suspension of RBCs was washed with HBSS twice. When indicated, RBCs were stressed (sRBCs) by heating for 30 min at 48 °C with shaking. In total, $1 \times 10^{10}$ RBC were resuspended in 1 ml diluent C, mixed with 1 ml diluent C containing 7.5 µM PKH26 (Sigma-Aldrich, #MINI26-1KT), and incubated in the dark for 5 min at 37 °C. The reaction was stopped by adding 2 ml HBSS/ 1% BSA, and the suspension was washed with HBSS. For in vivo experiments, RBCs were resuspended to 50% hematocrit in HBSS. RBC ghosts were obtained by lysis of stressed PKH26-stained RBCs derived from C57BL/6J mice (volume of the suspension was two times more than PKH26-stained UBI-GFP/BL6 RBCs) with PBS to water (1:15) mixture, followed by at least three centrifugations at 13,000× *g*, 5 min each, until the pellet became brighter. The final pellet was resuspended in HBSS.

## Immunofluorescence

The mouse liver was fixed in 4% PFA at 4 °C for 24 h. Tissues were washed with PBS (3 × 30 min) and soaked in 12.5% and 25% sucrose for 1.5 h and 48 h, respectively. Liver samples were embedded in Cryomatrix medium (Epredia™, #6769006), frozen in liquid nitrogen, sectioned in 10-µm slices using a cryo-microtome (Leica), and stored at –20 °C. Before staining, the sections were left in RT for 20 min, surrounded with Pap Pen Liquid Blocker (Ted Pella, 22311), washed in PBS for 10 min, and permeabilized with PBS/0.1% Triton X-100 (Thermo Fisher, #85111) for 20 min. Non-specific antibody binding was blocked by incubating sections in PBS/3% BSA (Bioshop, #ALB001.250) for 2 h at RT. For LSEC detection, tissues were incubated with anti-CD146 PE (BioLegend, #134704) 1:100 in PBS/3% BSA for 2 h at RT. For KCs detection, tissues were incubated with anti-F4/80 PE (BioLegend, #123110) 1:100 in PBS/3% BSA for 2 h at RT. After antibody incubation, tissues were washed 5 × 5 min with PBS/0.1% Triton X-100 to remove unbound antibodies. For nuclear staining, sections were incubated with Hoechst 33342 (ThermoFisher, #H3570) diluted in PBS/0.1% Triton X-100 (final concentration— 1 µg/ml) for 10 min at RT. Next, sections were washed 3×5 min with PBS/0.1% Triton X-100, incubated in PBS for 10 min, and mounted with ProLong™ Glass Antifade Mountant (Thermo Fisher, #P36982). For immunofluorescence staining of in vitro cultured NPCs, cells were seeded on round coverslips coated with collagen I (Corning, #354236). For macropinosomes visualization, seeding of NPCs was performed with an additional step of macrophage depletion with anti-F4/80 magnetic beads (Miltenyibiotec, #130-110-443) according to the manufacturer's protocol. For Hb and dextran intracellular colocalization, cells were fixed with 4% PFA for 10 min at RT, washed with PBS, and incubated in PBS/5% BSA for 1 h at RT for blocking. After blocking, cells were incubated overnight at 4 °C with anti-F4/80 PE and/or anti-STAB2 (St John's Laboratory, #STJ192359, 1:100). After incubation with primary antibody, cells were washed with PBS, and incubated with secondary antibody: donkey-anti-rabbit IgG AlexaFluor 488 (Thermo Fisher Scientific, #A-31571, 1:300) for 1 h at RT. For FPN staining, NPCs were washed with PBS and blocked with 1% BSA/PBS for 1 h at RT. After blocking, cells were stained with anti-CD31 FITC (Biolegend, #102506, 1:50), anti-F4/80 PE (Biolegend, #123110, 1:100), and anti-FPN (Kind gift from Tara Arvedson, Amgen, 1:100 -conjugated with AF647 fluorescent dye—Abcam, #ab269823) for 45 min at RT. NPCs were washed with PBS and fixed with 4% PFA for 10 min at RT. After fixation, cells were washed with PBS, stained with Hoechst 33342 for 10 min at RT, and mounted on the slides with ProLong™ Glass Antifade Mountant.

For intracellular HO1 staining, NPCs were washed with PBS and blocked with 1%BSA/PBS for 1 h at RT. After blocking, cells were stained with anti-CD31 FITC (Biolegend, #102506, 1:50) and anti-F4/80 PE (Biolegend, #123110, 1:100), washed with PBS, and fixed with 4% PFA for 10 min at RT. After fixation, NPCs were washed with PBS, permeabilized with PBS/0.1% Triton X-100 for 5 min at RT, and blocked again with 1%BSA/PBS for 1 h at RT. After blocking, cells were incubated overnight at 4 °C with anti-HO1 (Enzo, #ADI-OSA-150-F, 1:100) primary antibody. After incubation, cells were washed and incubated with donkey anti-rabbit IgG

AlexaFluor 647 (Thermo Fisher Scientific, #A-31571, 1:300) for 1 h at RT. NPCs were washed, stained with Hoechst 33342 for 10 min at RT, and mounted on the slides with ProLong™ Glass Antifade Mountant. Samples were imaged using a confocal microscope LSM 800 (Zeiss), equipped with an EC Plan-Neofluar 40x/1.30 Oil DIC M27 oil objective and T-PMT detectors. Images were acquired with a resolution of 2101 ×2101 pixels. The images were processed using ImageJ software with linear adjustments of contrast and brightness, equally across the whole image area and in comparison to the blank samples. For macropinosomes visualization, fixed cells were permeabilized and blocked with solution I—PBS/ 0,1% w/v Saponin, 0,2% w/v/gelatin, 5 mg/ml BSA for 10 min RT. After blocking, cells were incubated in solution II—PBS 0.01% Saponin, 0,2% gelatin with primary antibody anti-EEA1 (Enzo Life Sciences, #ALX-210-239, 1:1000) for 1 h at RT. After washing in solution II, cells were incubated with secondary antibody donkey anti-rabbit IgG 488 (Thermo Fisher Scientific, #A-21206, 1:500) for 1 h at RT. Phalloidin Atto 390 (Sigma-Aldrich, #50556, 1:500) to stain actin was added during incubation with fluorescent secondary antibodies. After incubation with antibodies, cells were washed with PBS and mounted on the slides with Moviol. Sections and slides were visualized with a Zeiss LSM 710 equipped with an Objective Plan-Apochromat 63×/1.4 Oil DIC M27 oil objective and T-PMT detectors. Macropinosome size was determined with ZEN 2011 software. The remaining images were processed using Adobe Photoshop software with linear adjustments of contrast and brightness.

Human non-parenchymal cells seeded on collagen I-coated plates (Ibidi, #80841) were washed with PBS and fixed with 4% PFA for 15 min at RT. After fixation, cells were washed with PBS and blocked in 2% BSA for 1 h. After blocking, cells were incubated with anti-human CD36 PE (Biolegend, #336206, 1:50), anti-human CD32B PE (Biolegend, #398404, 1:50), or anti-human CD36 BUV421 (Biolegend, # 336230, 1:50) antibodies for 1 h. For intracellular EEA1 staining, cells were additionally permeabilized and blocked with PBS/0.1% w/v Saponin, 0.2% w/v/gelatin, 5 mg/ml BSA solution for 10 min RT. After blocking, cells were incubated in PBS 0.01% Saponin +0.2% gelatin solution with primary antibody anti-EEA1 (Enzo Life Sciences, #ALX-210-239, 1:1000) for 1 h at RT. After washing, cells were incubated with secondary antibody donkey anti-rabbit IgG 488 (Thermo Fisher Scientific, #A-21206, 1:500) for 1 h at RT, washed with PBS and directly visualized using a confocal microscope LSM 800 (Zeiss).

## Proteomics

Cell pellets were lysed in lysis buffer (2% SDS, 100 mM TRIS, pH 8.5). Samples were subjected to chloroform/methanol precipitation, and the resulting protein pellets were washed twice with methanol, air-dried, and resuspended in 100 mM HEPES (pH 8) by vigorous vortexing and sonication. Proteins were reduced and alkylated with 10 mM TCEP/40 mM CAA and digested with trypsin in a 1:100 (w/w) enzyme-to-protein ratio at 37 °C overnight. Digestion was stopped by the addition of trifluoroacetic acid (TFA) to a final concentration of 1%, and peptides were desalted on C18-StageTips. Chromatographic separation was performed on an UltiMate 3000 nano-LC system (Thermo Fisher Scientific) coupled to a Q Exactive HF-X mass spectrometer via an Easy-Spray source. The Q Exactive HF-X was operated in data-dependent mode, and MS data were processed with MaxQuant v.

1.6.17.0 (Cox and Mann, 2008). Peptides were identified against the mouse reference proteome UP000000589 using the Andromeda search engine. Cysteine carbamidomethylation was set as a fixed modification, and methionine oxidation, glutamine/asparagine deamidation, and protein N-terminal acetylation were set as variable modifications. For in silico digests of the reference proteome, cleavages of arginine or lysine followed by any amino acid were allowed (trypsin/P), and up to two missed cleavages were allowed. False discovery rate (FDR) was set to 0.01 at the peptide, protein, and site levels. Match between runs was enabled. Quantification was based on unique and razor peptides. Processed data were imported into Perseus v. 1.6.10.0 (Tyanova et al, 2016) for downstream statistical analysis.

For the absolute quantification of protein abundance in LSECs compared to splenic and heart ECs and iron-recycling macrophages, peptides were separated on an Easy-Spray Acclaim PepMap column (2 μm, 100 Å, 75 μm × 50 cm) with a 180 min acetonitrile gradient at a flow rate of 300 nl/min. Survey scans were acquired at a resolution of 120,000 (m/z 200), with MS/MS spectra collected at 15,000 resolution. The top 12 most abundant ions (charge states 2–5) were isolated with a 1.3 m/z window and fragmented by HCD (27 NCE), with a dynamic exclusion of 40 s. Maximum ion injection times were 45 ms (MS) and 96 ms (MS/MS), with target values of 3e6 and 1e5, respectively; the MS/MS intensity threshold was set to 1e4. Both LFQ and iBAQ values were log2-transformed in Perseus. Standard filtering was applied to remove reverse hits, contaminants, and proteins only identified by site. For relative comparisons between LSEC and other cell types, a minimum of two LFQ values per group was required; missing values were imputed from a normal distribution (downshift = 1.8 SD, width = 0.3 SD). Two-sided Student's $t$ tests (permutation-based FDR = 0.01, S0 = 0.2) identified proteins significantly altered between groups. Data are reported in Dataset EV1.

For the comparison of proteomic changes in LSECs upon stressed RBCs injection in mice, peptides were separated on an Easy-Spray Acclaim PepMap column (2 μm, 100 Å, 50 μm × 15 cm) with a 60 min acetonitrile gradient at a flow rate of 300 nl/min. Survey scans were acquired at a resolution of 60,000 (m/z 200), with MS/MS spectra collected at 15,000 resolution. The top 12 most abundant ions (charge states 2–5) were isolated with a 1.3 m/z window and fragmented by HCD (27 NCE), with a dynamic exclusion of 30 s. Maximum ion injection times were 50 ms (MS) and 28 ms (MS/MS), with target values of 3e6 and 1e5, respectively; the MS/MS intensity threshold was set to 3.6e4. LFQ intensity values were log2-transformed, and filtering was applied as above. Two-sided Student's $t$ tests (permutation-based FDR = 0.05, S0 = 0.2) identified proteins significantly altered between RBC-injected and control samples. Data are reported in Dataset EV2.

## Heme analysis and iron quantification

Tissue non-heme iron content was measured using the bathophenanthroline colorimetric method and calculated against dry tissue weight. Serum iron content (SFBC) and unsaturated iron-binding capacity (UIBC) were measured with the SFBC and the UIBC kit (Biolabo, #97408). Transferrin saturation was calculated using the formula SFBC/(SFBC + UIBC) × 100. Heme content was measured as described previously (Slusarczyk et al, 2023). Briefly, to determine the extracellular heme content in the spleen, the entire

spleen was dissected, weighed, and gently mashed through a 100 μm strainer, rinsed with 3 ml of HBSS (ThermoScientific, #14025092). The resulting suspension was centrifuged at 400× g for 10 min at 4 °C, and the supernatant was collected. To eliminate the remaining cells and membranes, the supernatant was centrifuged again at 1000× g for 10 min at 4 °C, and the supernatant was transferred to a fresh tube. To determine heme content in the kidneys, organs were incubated in RIPA buffer containing Protease Inhibitor Cocktail, followed by centrifugation for 15 min at 12,000× g. The obtained lysate was further used for heme quantification. To determine heme content in the blood, blood from the circulation and portal vein was collected into the heparin-coated tube, centrifuged at 400× g for 10 min at 4 °C, and plasma was transferred to a fresh tube. Heme concentrations in the plasma, kidney, or supernatant collected from magnetically sorted LSECs, KCs, and RPMs were determined using the Heme Assay Kit (Sigma-Aldrich, #MAK316), following the instructions provided by the manufacturer. The absorbance was measured at a wavelength of 400 nm. The amount of heme was calculated against Heme Calibrator and additionally normalized to the initial weight of fresh spleens (for splenic extracellular heme content) or protein concentration (for kidney and magnetically sorted cells).

## RNA isolation and qPCR

RNA from tissues was extracted using TRIzol™ Reagent (Thermo-Scientific, #15596026) and from sorted cells using TRIzol™ LS Reagent (ThermoScientific, #10296028) according to the manufacturer's instructions with an additional step of RNA precipitation with glycogen (ThermoScientific, #R0551). RNA (100–400 ng) was reverse transcribed into cDNA using random primers (ThermoScientific, #48190011) and RevertAid H Minus Reverse Transcriptase (Thermo-Scientific, #EP0452). The cDNA products were amplified by quantitative polymerase chain reaction (qPCR) using the SG qPCR Master Mix (2×) (Eurex, #E0401-01). Real-time qPCR (RT-qPCR) was run in LightCycler 96 Real-Time PCR System (Roche), using the following primer sequences: *Rpl19*- AGGCATATGGGCATAGGG AAGAG (forward primer); TTGACCTTCAGGTACAGGCTGTG (reverse primer). *Bmp6* – ATGGCAGGACTGGATCATTGC (for-ward primer); CCATCACAGTAGTTGGCAGCG (reverse primer). *Hmox-1* – AGGCTAAGACCGCCTTCCT (forward primer); TGTGTTCCT CTGTCAGCATCA (reverse primer). *Hamp* – ATACC AATGCAGAAGAGAAGG (forward primer); AACAGATACCAC ACTGGGAA (reverse primer).

## RNA sequencing

Transcriptome analysis of LSECs was conducted using the AmpliSeq method. RNA integrity was assessed using Agilent RNA 6000 Nano Kit on Agilent™ 2100 Bioanalyzer™, and cDNA library preparation was performed using a commercially available kit (Ion AmpliSeq™ Transcriptome Mouse Gene Expression Kit, ThermoScientific, # A36554) following the manufacturer's recom-mendations. Targeted cDNA fragments were amplified using the Ion AmpliSeq™ Library Kit 2.0. cDNA libraries were measured using Agilent™ 2100 Bioanalyzer™. Raw reads were processed using the Torrent Suite tool and mapped to the mouse genome mm10 AmpliSeqTranscriptome version using Torrent Mapping Align-ment Program (TMAP).

## Elemental analysis by inductively coupled plasma optical emission spectrometry (ICP-OES) of liver non-parenchymal cells and red pulp macrophages

Samples were digested overnight in 65% ultrapure nitric acid (Merck, #1004410250) and diluted to a final volume of 5 mL before measuring the metal content on a iC at 0.5 L/min, and nebulizer gas pressure at 2.03 bar AP Pro ICP-OES (Thermo-Fisher Scientifics) instrument equipped with a iSC-65 autosampler, a Torch Duo (Slot) Rev 02 argon plasma torch and a Qutegra software, which was used to calculate the metal concentration in measured sample. The wavelengths for Fe (259.940 nm) and S (180.731 nm) were selected to minimize overlapping emissions. Samples were measured in Aqueous-Axial_iFR mode with RF power set to 1250 W, nebulizer gas flow at 0.5 L/min, and nebulizer gas pressure at 2.03 bar. Final metal concentration was calculated according to the dilution of the sample and normalized to the protein level/number of cells used for digestion.

## Graphics

The synopsis image and graphical abstract were created using BioRender.com.

## Statistical analysis

Statistical analyses were conducted using GraphPad Prism software (Version 9.4.1). Data are represented as the mean ± SEM. The number of mice/samples per group or the number of independent cell-based experiments is shown in the figures and reflects biological replicates. Sample size was determined based on power analysis, prior experience of performing similar experiments, and previously published papers in the field. Although no formal tests for normality or homogeneity of variance were performed, sample sizes and distributions were consistent with standard practice in the field. Statistical tests were chosen according to data type. When two groups were compared, a two-tailed unpaired Welch's $t$ test was applied, whereas for comparisons involving more than two groups, one-way or two-way analysis of variance (ANOVA) was performed, depending on the experimental design. Post-hoc Tukey's test was used to correct for multiple comparisons between groups. Results were considered significant for $P < 0.05$; the exact $P$ values are shown in the figures.

## Ethics declarations

All animal experiments were conducted following the guidelines of European Directive 2010/63/EU and the Federation for Laboratory Animal Science Associations. The procedures were approved by the Local Ethics Committee in Warsaw (No. WAW2/150/2019, WAW2/138/2019, WAW2/137/2020, WAW2/137/2021, WAW2/179/2021, WAW2/094/2021 and WAW2/067/2025), or for the dietary experiment by the local ethical committee in Olsztyn No. 26/2018. For the use of human cells, this study conformed to the principles set out in the WMA Declaration of Helsinki and the Department of Health and Human Services Belmont Report. It received approval from the Ethics Committee of the Faculty of

Medicine at Leipzig University (Biobank of Surgical Research, registration no. 322/17-ek, 2020/06/10, ratified on 2021/11/30; "Role of human LSEC in hemoglobin clearance", registration no. 057/23-ek, 2023/08/21).

## Data availability

Transcriptome analysis of LSECs was conducted using the AmpliSeq method, and proteomics was performed by label-free quantification with mass spectrometry as described in the Methods. LSEC transcriptomic data for iron citrate injection have been previously deposited in the GEO repository under accession no. GSE235976, whereas data for Hb injection are accessible under no. GSE240270. Proteomics data were deposited to the ProteomeXchange Consortium via the PRIDE partner repository with the dataset identifier PXD051274. Raw flow cytometry data for Figs. 3 and 5A-D and M are deposited in the BioStudies repository under the following links: Fig. 3A and C: ; Fig. 3E and G: https://www.ebi. ac.uk/biostudies/studies/S-BSST2282; Fig. 5A: https://www.ebi.ac.uk/ biostudies/studies/S-BSST2289; Fig. 5B: https://www.ebi.ac.uk/biostudies/ studies/S-BSST2286; Fig. 5C: https://www.ebi.ac.uk/biostudies/studies/S-BSST2294; Fig. 5D: https://www.ebi.ac.uk/biostudies/studies/S-BSST2293; Fig. 5M: https://www.ebi.ac.uk/biostudies/studies/S-BSST2295.

The source data of this paper are collected in the following database record: biostudies:S-SCDT-10_1038-S44319-025-00673-5.

## Peer review information

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

## Acknowledgements

This work was funded by the National Science Centre grant 2018/31/B/NZ4/03676 to KMS and the Foundation of Polish Science grant TEAM TECH/2016-1/8 to TPR. Additionally, TPR and MS were supported by the National Centre for Research grant 2020/39/B/NZ7/01382. Proteomic measurements were performed at the Proteomics Core Facility, IMol Polish Academy of Sciences, utilizing the equipment funded by the 'Regenerative Mechanisms for Health' project MAB/2017/2 within the International Research Agendas program of the Foundation for Polish Science, co-financed by the European Union under the European Regional Development Fund. We express gratitude to Marta Niklewicz for her substantial technical assistance in this project. We would like to thank Daria Zdżalik-Bielecka for her valuable advice during the initial phases of the project. Additionally, we extend our thanks to Damian Strzemecki for his technical assistance in performing mouse injections. We thank Tara Arvedson (Amgen Inc. USA) for the anti-ferroportin antibody, Elizabeta Nemeth for PR73, and Aneta Suwińska for sharing UBI-GFP/BL6 mice. We thank Julian Connor Eckel for his excellent technical assistance in the isolation and provision of primary human liver cells. Many thanks to Agnieszka Popielska and Anna Kosson, and the staff of the Experimental Medicine Centre (Bialystok, Poland) and Mossakowski Medical Research Institute (Warsaw, Poland) for their technical support. This research was supported by IIMCB IN-MOL-CELL Infrastructure (particularly Rodent Breeding and Microscopy Facilities) funded by the European Union-NextGenerationEU under the National Recovery and Resilience Plan. IN-MOL-CELL Infrastructure was also funded by the European Union under Horizon Europe (Project 101059801 - RACE) and by the RACE-PRIME project (FENG.02.01-IP.05-T003/23) carried out within the IRAP program of the Foundation for Polish Science, co-financed by the European Union under the European Funds for Smart Economy 2021-2027 (FENG).

## Author contributions

**Gabriela Zurawska**: Conceptualization; Formal analysis; Validation; Investigation; Visualization; Writing—original draft; Writing—review and editing. **Zuzanna Sas**: Conceptualization; Formal analysis; Validation; Investigation; Visualization; Writing—original draft. **Aneta Jończy**: Conceptualization; Formal analysis; Validation; Investigation; Visualization; Writing—original draft; Writing—review and editing. **Raghunandan Mahadeva**: Formal analysis; Investigation. **Patryk Slusarczyk**: Formal analysis; Investigation; Visualization. **Marta Chwałek**: Formal analysis; Investigation; Visualization. **Daniel Seehofer**: Resources; Methodology. **Georg Damm**: Resources; Methodology. **Rafał Mazgaj**: Methodology. **Marcin Skórzyński**: Methodology. **Maria Kulecka**: Formal analysis. **Izabela Rumieńczyk**: Formal analysis; Investigation. **Morgane Moulin**: Investigation. **Kamil Jastrzębski**: Investigation; Visualization. **Kevin Waldron**: Methodology. **Michal Mikula**: Data curation; Visualization. **Anders Etzerodt**: Resources; Writing—review and editing. **Remigiusz Serwa**: Data curation; Formal analysis; Validation; Methodology. **Marta Miączyńska**: Conceptualization; Formal analysis; Supervision; Writing—review and editing. **Tomasz P Rygiel**: Conceptualization; Formal analysis; Supervision; Funding acquisition; Validation; Investigation; Visualization; Writing—original draft; Project administration; Writing—review and editing. **Katarzyna Mleczko-Sanecka**: Conceptualization; Formal analysis; Supervision; Funding acquisition; Validation; Investigation; Visualization; Writing—original draft; Project administration; Writing—review and editing.

Source data underlying figure panels in this paper may have individual authorship assigned. Where available, figure panel/source data authorship is listed in the following database record: biostudies:S-SCDT-10_1038-S44319-025-00673-5.

## Disclosure and competing interests statement

The authors declare no competing interests.

# Expanded View Figures

**Figure EV1. LSECs represent the major cell type that sequesters Hb.**

(**A, B**) Hemoglobin (Hb) distribution in control and macrophage-depleted mice (clodronate) injected with AlexaFluor 750 labeled Hb (Hb-AF750, 10 μg/mouse), imaged with Bruker in vivo Imaging System. (**A**) The efficiency of macrophage depletion in the liver was examined by the percentage of liver KCs in control and clodronate-injected mice. (**B**) Representative images of organs isolated from Hb-AF750 (10 μg/mouse, 1 h) i.v.-injected mice. (**C**) Frozen liver slices from mice injected with Hb-AF647 (red) were processed and stained for Kupffer cells (KCs) (F4/80, green) or LSECs (CD146, green) and nuclei (blue). Merged-channel images for this figure are presented in Fig. 1C. (**D**) Murine NPCs in vitro cultures were treated with Hb-AF750 (0.5 μg/ml) for 1 h. Normalized Hb-AF750 fluorescence intensity and percentage of Hb-AF750+ LSECs and KCs, measured with flow cytometry. (**E**) Plots illustrating high mRNA expression of human LCES markers CD32B and CD36 in liver ECs, visualized using the Human Liver Cell Atlas (Guilliams et al, 2022). Data are expressed as mean ± SEM, and each data point represents one biological replicate, $n = 10$ (**A**), 4 (**D**). Welch's unpaired $t$ test was used to determine statistical significance in (**A, D**); exact $P$ values are shown on graphs.

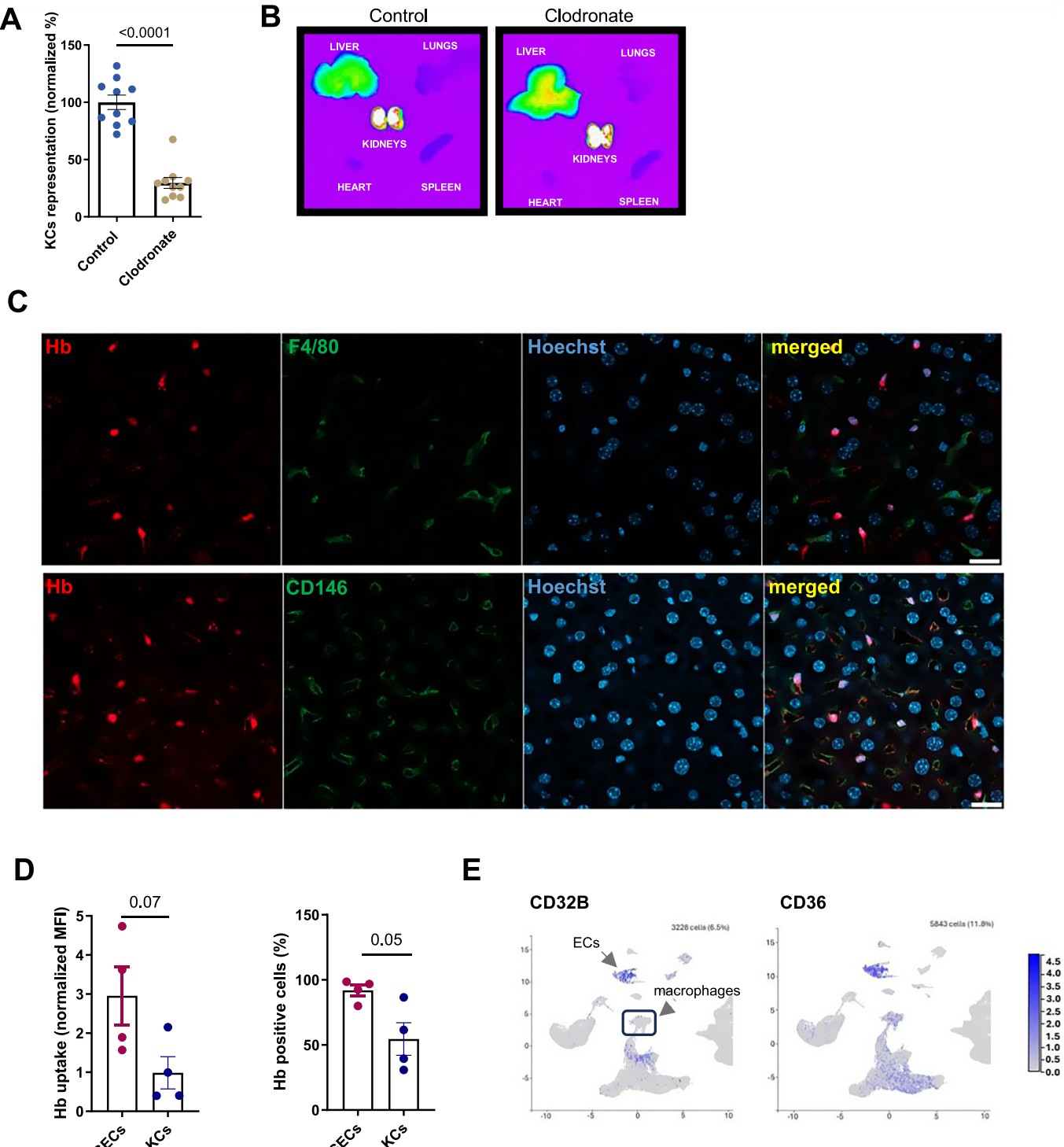

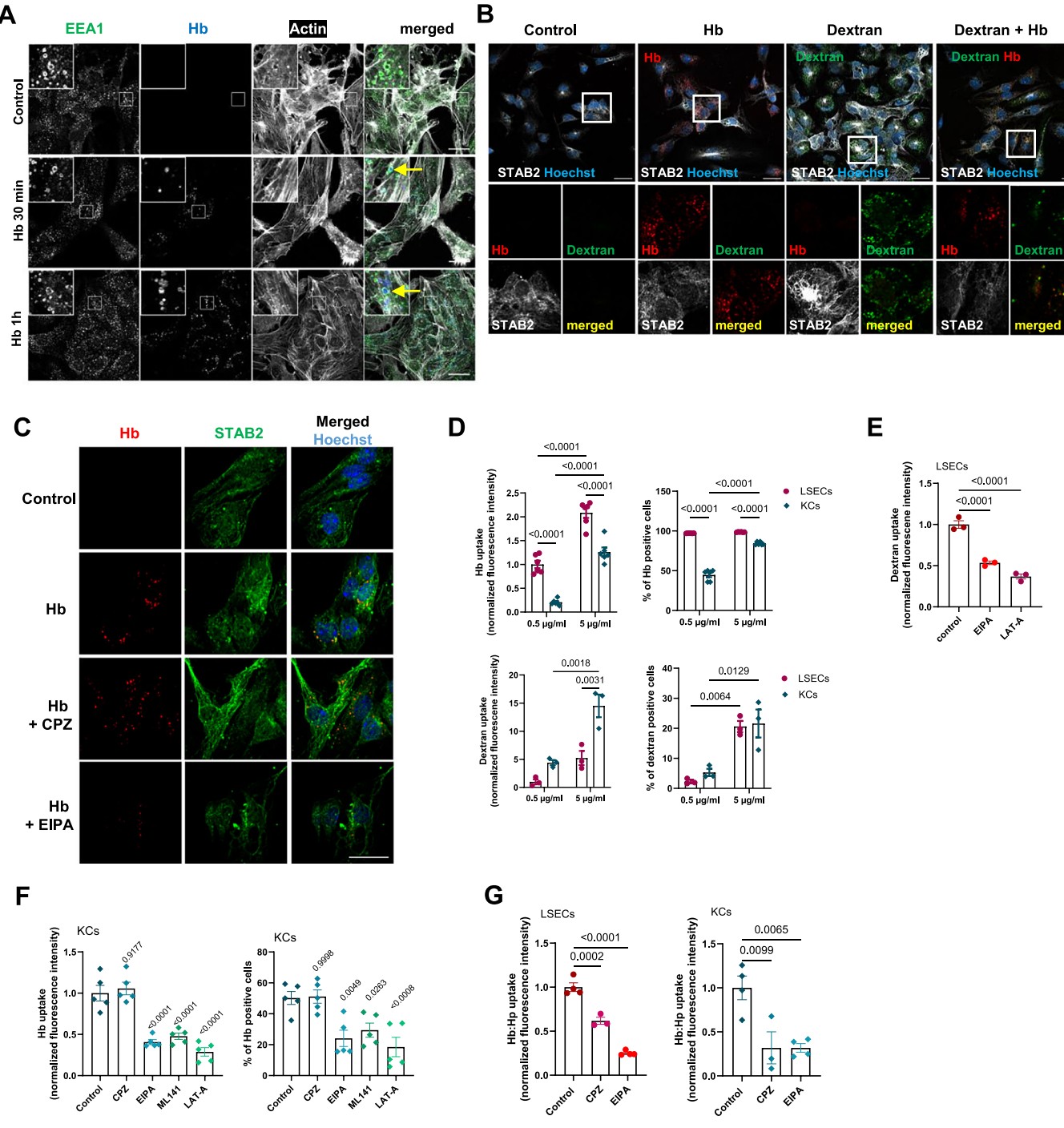

**Figure EV2. Primary LSECs and KCs take up Hb via macropinocytosis.**

Murine NPCs in vitro cultures were treated with Hb-AF647 or fluorescently-labeled dextran (0.5 or 5 µg/ml) for 10 min, 30 min, or 1 h. (A) Hb-AF647 vesicle localization was imaged in NPCs in vitro cultures depleted of macrophages. Arrows indicate Hb-AF647 (blue) presence in the EEA-1+ (green) and Actin+ (white) vesicles. Areas in the highlighted rectangles are shown at higher magnification (left upper corner). Scale bars, 20 µm. Merged-channel images for this figure are presented in Fig. 2C. (B) Co-localization of Hb-AF647 (red) with rhodamine-dextran (green) was imaged in STAB2+ LSECs (white). The arrow indicates co-localization of Hb and rhodamine-dextran at the 10 min time-point. The area in the highlighted rectangle is shown at higher magnification in separate channels below. Nuclei were stained with Hoechst (blue). Scale bar, 20 µm. Merged-channel images for this figure are presented in Fig. 2D. (C) NPCs were pretreated with the inhibitor of clathrin-mediated endocytosis chlorpromazine (CPZ, 2 µM) or the macropinocytosis blocker EIPA (25 µM), before Hb-AF647 treatment for 10 min. Cells were fixed and stained for STAB2 (green) and nuclei (Hoechst, blue). Scale bars, 20 µm. (D–F) Hb-AF750/FITC-dextran fluorescence intensity and percentage of Hb-AF750 + /dextran+ LSECs or KCs, measured with flow cytometry 1 h after (D) administration of the increasing doses of the indicated cargo or (E, F) upon pretreatment with the indicated inhibitors. (G) Hb:Hp-AF750 fluorescence intensity in LSECs or KCs upon pretreatment with the indicated inhibitors, measured with flow cytometry 1 h after administration of Hb:Hp-AF750. Numerical data are expressed as mean ± SEM, and each data point represents one biological replicate, $n = 3$–6 (D), 3 (E), 5 (F), 3–4 (G). Two-way ANOVA with Tukey's Multiple Comparison test was used to determine statistical significance in (D); one-way ANOVA with Tukey's Multiple Comparison test was used to determine statistical significance in (E–G); exact $P$ values are shown on graphs, values above bars in (F) indicate comparison with control cells. Source data are available online for this figure.

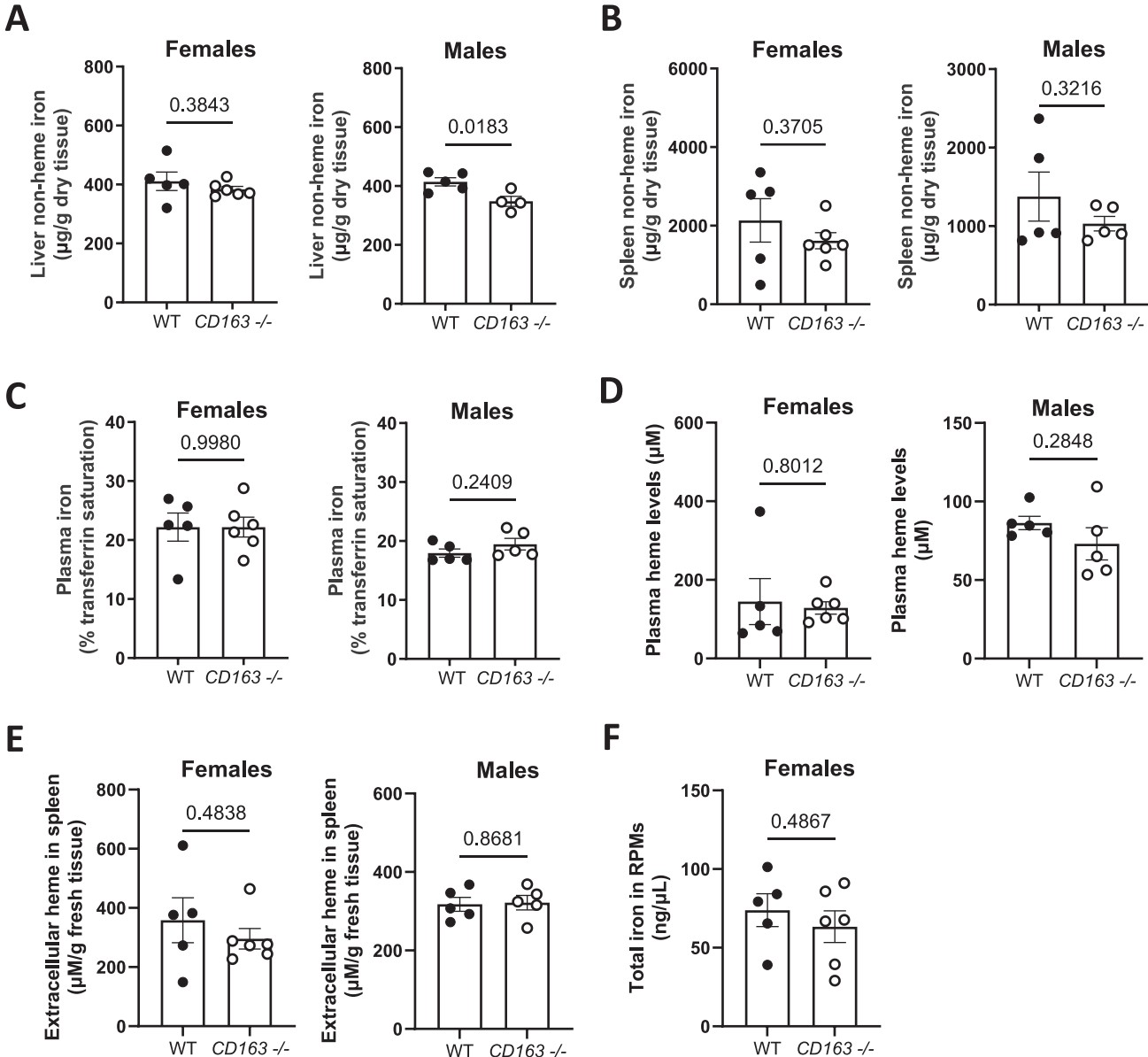

**Figure EV3. CD163 KO mice show no major differences in systemic and splenic iron parameters.**

(A–E) The phenotype of Cd163−/− mice was compared with wild-type (WT) littermates. (A, B) Non-heme iron content in (A) the liver and (B) spleen of female and male mice. (C) Plasma iron levels were determined by transferrin saturation measurements. (D, E) Heme levels were measured in the (D) plasma and (E) extracellular fluid from the spleen using Heme Assay Kit. (F) Total iron levels in magnetically-sorted RPMs were measured with Iron assay kit. Data are expressed as mean ± SEM, and each data point represents one biological replicate, $n = 4$–6 (A, C), 5–6 (B, D, F), 5 (E). Welch's unpaired $t$ test was used to determine statistical significance; exact $P$ values are shown on graphs.

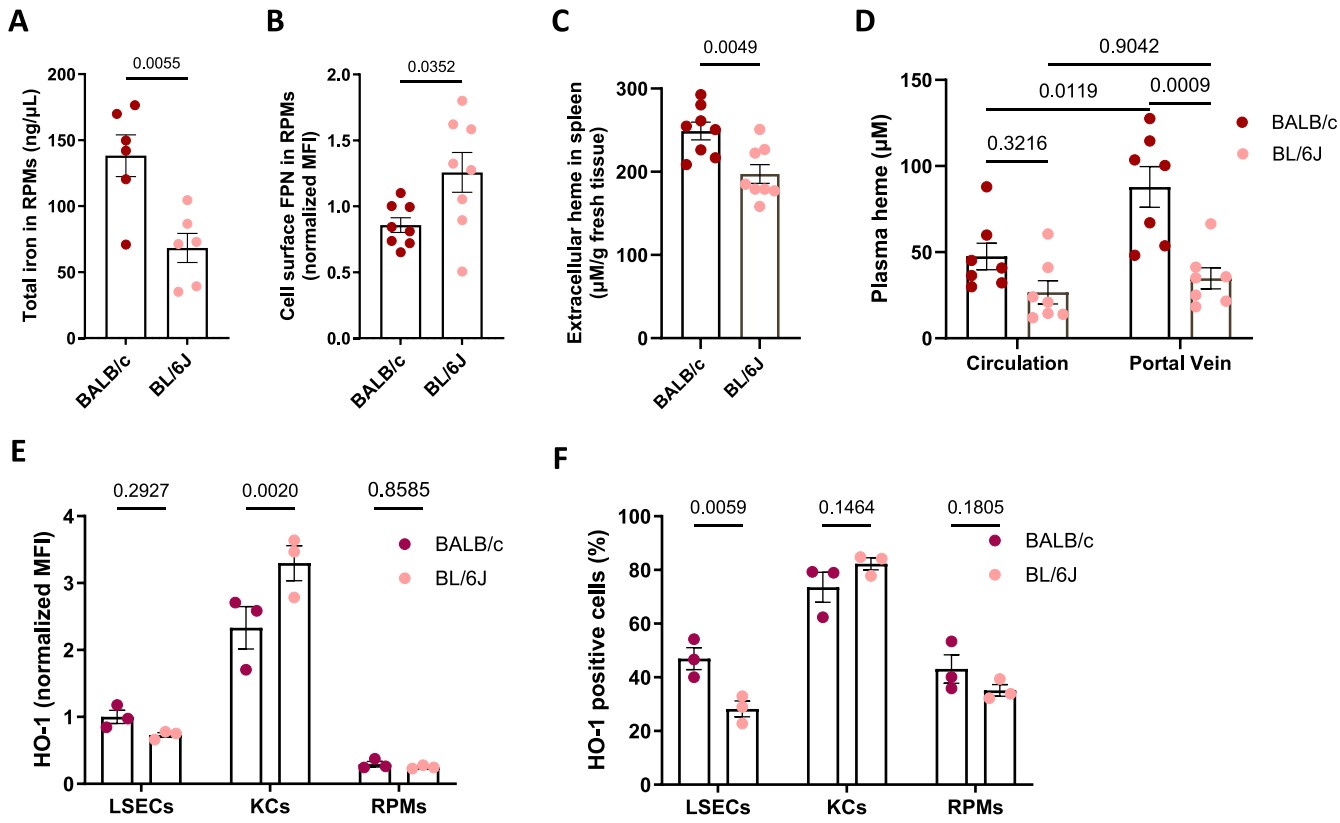

**Figure EV4. Comparison of iron-recycling parameters between BALB/c and C57BL/6J mice.**

(A) The total intracellular iron content in magnetically-sorted RPMs was assessed using the Iron Assay Kit. (B) FPN surface levels were measured in RPMs by flow cytometry. (C) Extracellular heme content in the spleen and (D) heme levels in the portal vein and circulating plasma were measured using Heme Assay Kit. (E, F) HO-1 levels and percentage of HO-1 positive cells in single cell suspensions from respective organs of BALB/c and C57BL/6J (BL/6J) were determined using flow cytometry. Data are expressed as mean ± SEM, and each data point represents one biological replicate, $n = 6$ (A), 8 (B, C), 7 (D), 3 (E, F). Welch's unpaired $t$ test was used to determine statistical significance in (A–C), while two-way ANOVA with Tukey's Multiple Comparison tests was used in (D–F); exact $P$ values are shown on graphs.

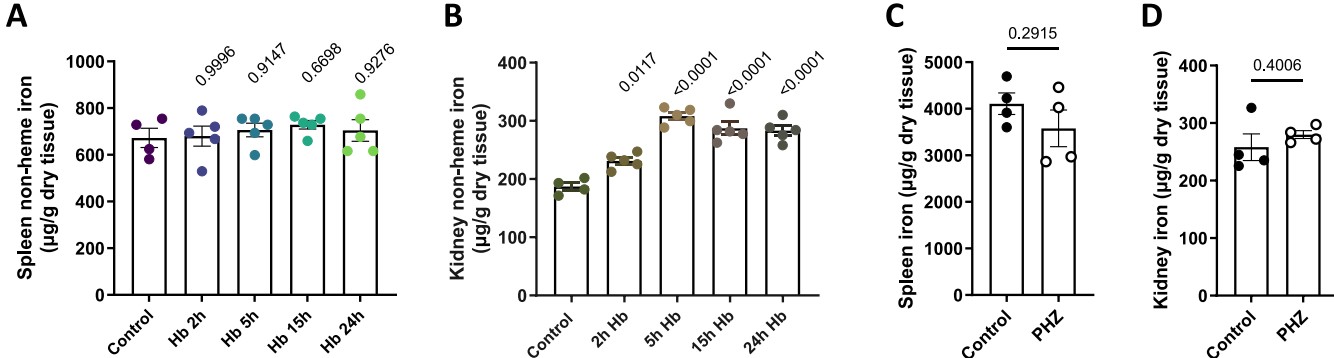

**Figure EV5. Alterations of splenic and renal iron levels upon Hb injection and PHZ-induced hemolysis.**

(**A**, **B**) Mice were injected with Hb (10 mg/mouse) for the indicated time points. Non-heme iron content in the (**A**) spleens and (**B**) kidneys. (**C**, **D**) Hemolysis was induced by i.p. injection of phenylhydrazine (PHZ, 0.125 mg/g) for 6 h. Non-heme iron content in the (**C**) spleens and (**D**) kidneys. Data are expressed as mean ± SEM, and each data point represents one biological replicate, $n = 4$–5 (**A**, **B**), 4 (**C**, **D**). Welch's unpaired $t$ test was used to determine statistical significance in (**C**, **D**). One-way ANOVA with Tukey's Multiple Comparison test was used in (**A**, **B**); exact $P$ values are shown on graphs, values above bars in (**A**, **B**) indicate comparison with control mice.

 