## [Peer Review File · EMBO Reports]

Liver sinusoidal endothelial cells constitute a major route for hemoglobin clearance

Gabriela Zurawska, Zuzanna Sas, Aneta Jonczy, Raghunandan Mahadeva, Patryk Slusarczyk, Marta Chwalek, Daniel Seehofer, Georg Damm, Rafał Mazgaj, Marcin Skórzyński, Maria Kulecka, Izabela Rumiencyk, Morgane Moulin, Kamil Jastrzebski, Kevin Waldron, Michał Mikula, Anders Etzerodt, Remigiusz Serwa, Marta Miaczynska, Tomasz Rygiel, and Katarzyna Mleczko-Sanecka

Corresponding author(s): Katarzyna Mleczko-Sanecka (kmsanecka@iimcb.gov.pl) , Tomasz Rygiel (trygiel@imdik.pan.pl)

Review Timeline:

Submission Date:	20th Feb 25
Editorial Decision:	20th Mar 25
Revision Received:	9th Oct 25
Editorial Decision:	7th Nov 25
Revision Received:	20th Nov 25
Accepted:	3rd Dec 25

Transaction Report:

Dear Dr. Mleczko-Sanecka

Thank you for the submission of your research manuscript to our journal. We have now received the full set of referee reports that is copied below.

As you will see, the referees acknowledge that the findings are interesting and that the conclusions are overall supported by the data presented but they also raise a few concerns and have suggestions how to further strengthen the data that should be addressed.

Given these constructive comments, we would like to invite you to revise your manuscript with the understanding that the referee concerns (as detailed above and in their reports) must be fully addressed and their suggestions taken on board. Please address all referee concerns in a complete point-by-point response. Acceptance of the manuscript will depend on a positive outcome of a second round of review. It is EMBO Reports policy to allow a single round of revision only and acceptance or rejection of the manuscript will therefore depend on the completeness of your responses included in the next, final version of the manuscript.

We realize that it is difficult to revise to a specific deadline. In the interest of protecting the conceptual advance provided by the work, we recommend a revision within 3 months (June 20). Please discuss the revision progress ahead of this time with the editor if you require more time to complete the revisions.

I am also happy to discuss the revision further via e-mail or a video call, if you wish.

=====
IMPORTANT NOTE:

We perform an initial quality control of all revised manuscripts before re-review. Your manuscript will FAIL this control and the handling will be delayed IN CASE the following APPLIES:

- 1) A data availability section providing access to data deposited in public databases including an URL that directly resolves to the dataset is missing. If you have not deposited any data, please add a sentence to the data availability section that explains that.
- 2) Your manuscript contains statistics and error bars based on $n=2$. Please use scatter blots in these cases. No statistics should be calculated if $n=2$.

=====

- 1) a .docx formatted version of the manuscript text (including legends for main figures, EV figures and tables). Please make sure that the changes are highlighted to be clearly visible.
- 2) individual production quality figure files as .eps, .tif, .jpg (one file per figure). Please download our Figure Preparation Guidelines (figure preparation pdf) from our Author Guidelines pages <https://www.embopress.org/page/journal/14693178/authorguide> for more info on how to prepare your figures.
- 3) a .docx formatted letter INCLUDING the reviewers' reports and your detailed point-by-point responses to their comments. As part of the EMBO Press transparent editorial process, the point-by-point response is part of the Review Process File (RPF), which will be published alongside your paper.
- 4) a complete author checklist, which you can download from our author guidelines (<<https://www.embopress.org/page/journal/14693178/authorguide>>). Please insert information in the checklist that is also

reflected in the manuscript. The completed author checklist will also be part of the RPF.

5) Please note that all corresponding authors are required to supply an ORCID ID for their name upon submission of a revised manuscript (<<https://orcid.org/>>). Please find instructions on how to link your ORCID ID to your account in our manuscript tracking system in our Author guidelines (<<https://www.embopress.org/page/journal/14693178/authorguide#authorshipguidelines>>)

6) We replaced Supplementary Information with Expanded View (EV) Figures and Tables that are collapsible/expandable online. A maximum of 5 EV Figures can be typeset. EV Figures should be cited as "Figure EV1, Figure EV2" etc... in the text and their respective legends should be included in the main text after the legends of regular figures.

7) Please format the "Data Availability" section at the end of the Methods using this (or similar) wording: "The [structural coordinates | microarray | mass spectrometry] data from this publication have been deposited to the [name of the database] database [URL] and assigned the identifier [accession | permalink | hashtag].". It is important to include an URL that resolves directly to the dataset, not just the database.

Additional information on source data and instruction on how to label the files are available <<https://www.embopress.org/page/journal/14693178/authorguide#sourcedata>>.

10) Figure legends and data quantification:
The following points must be specified in each figure legend:

- the name of the statistical test used to generate error bars and P values,
 - the EXACT p-values,
 - the number (n) of independent experiments (please specify technical or biological replicates) underlying each data point,
 - the nature of the bars and error bars (s.d., s.e.m.)
- If the data are obtained from n {less than or equal to} 5, show the individual data points in addition to the SD or SEM.
- If the data are obtained from n {less than or equal to} 2, use scatter blots showing the individual data points.

See also the guidelines for figure legend preparation:
<https://www.embopress.org/page/journal/14693178/authorguide#figureformat>

11) Our journal encourages inclusion of *data citations in the reference list* to directly cite datasets that were re-used and obtained from public databases. Data citations in the article text are distinct from normal bibliographical citations and should directly link to the database records from which the data can be accessed. In the main text, data citations are formatted as follows: "Data ref: Smith et al, 2001" or "Data ref: NCBI Sequence Read Archive PRJNA342805, 2017". In the Reference list, data citations must be labeled with "[DATASET]". A data reference must provide the database name, accession number/identifiers and a resolvable link to the landing page from which the data can be accessed at the end of the reference. Further instructions are available at <<https://www.embopress.org/page/journal/14693178/authorguide#referencesformat>>.

12) Materials and Methods must be part of the main manuscript. We do not allow the inclusion of methods in the Supplement.

All Materials and Methods need to be described in the main text using our 'Structured Methods' format. According to this format, the Methods section includes a Reagents and Tools Table (listing key reagents, experimental models, software and relevant equipment and including their sources and relevant identifiers) followed by a Methods and Protocols section describing the methods, ideally using a step-by-step protocol format. The aim is to facilitate adoption of the methodologies across labs. Please download and fill our Reagents and Tools Table template (.docx), which you can find in our author guidelines: <https://www.embopress.org/page/journal/14693178/authorguide#structuredmethods>.

When submitting your revised manuscript, please do NOT include the Reagents and Tools Table in the Methods section of the manuscript but upload it as a separate file choosing the file type "Reagent Table".

An example of a Method paper with Structured Methods can be found here: <https://www.embopress.org/doi/10.15252/msb.20178071>.

13) As part of the EMBO publication's Transparent Editorial Process, EMBO Reports publishes online a Review Process File to accompany accepted manuscripts. This File will be published in conjunction with your paper and will include the referee reports, your point-by-point response and all pertinent correspondence relating to the manuscript.

Yours sincerely,

=====

Referee #1:

This is a very extensively documented, comprehensive analysis of the role of liver sinusoidal endothelial cells (LSEC) in the catabolism of hemoglobin generated during hemolysis. The manuscript successfully challenges the existing paradigm that attributes the erythrocyte-degrading function exclusively to macrophages. The authors convincingly demonstrate a novel pathway for the recycling of material from damaged erythrocytes wherein the spleen releases free hemoglobin and erythrocyte ghosts to the liver. In the liver, LSECs take up the free hemoglobin, and Kupffer cells (macrophages) take up the red cell ghosts. The authors extensively document the specialization of LSECs in hemoglobin catabolism and their dominant role in handling free hemoglobin releases by hemolysis. They also show the effect of hemoglobin uptake by LSECs on iron regulation. Hemoglobin uptake induces secretion of the critical iron regulator BMP6 from LSECs. BMP6 then induces hepcidin secretion from hepatocytes, allowing hemoglobin produced by hemolysis to influence systemic iron metabolism. Overall, this is a tour de force contribution to our understanding of the coordinated response of the organism to hemolysis, with implications for a variety of important pathophysiological settings including hemolytic anemias and transfusion therapy, and possibly also normal erythrocyte turnover.

This will be an influential contribution to the field of erythrocyte biology and iron metabolism.

My suggestions are minor:

1) Lines 120-125 should be reworded: "A sterile aqueous iron citrate (FeCit, 150 µg/mouse) solution (Sigma-Aldrich, #F3388) or sterile citric acid buffer (0.05 M, Sigma-Aldrich, #251275) was normalized to pH 7.0 and administered i.v. for 5 h. Mini-hepcidin (PR73, 50 nmol/mouse) (kind gift from Elizabeta Nemeth, UCLA, USA) was injected intraperitoneally (i.p.) for 4 h. To induce hemolysis, a sterile solution of phenylhydrazine (PHZ) (Sigma-Aldrich, #P26252) in PBS was administered i.p. at a dose of 0.125 mg/g of body weight for 6 h."

Reword to indicate that tissues were harvest X number of hours after the administration of substances, rather than imply that the substances were administered for X hours.

- 2) Figure 4H and line 286: The authors do not statistically analyze the difference between the responses of LSEC and KC to PR73 so they cannot make comparisons other than that the responses appear similar.
- 3) Figure 4A, FPN panel: It is hard to see what the arrows are pointing to. The related CD146 and F4/80 panels also appear dim.

Referee #2:

This study provides evidence that liver sinusoidal endothelial cells (LSECs) have a major role in clearance of free hemoglobin (Hb) from lysed red blood cells (RBCs). They take up Hb via micropinocytosis and express iron-handling proteins, including the heme-degrading enzyme Hmox1. Previous data suggested that there is some degree of hemolysis during physiological erythrophagocytosis in the spleen. Based on this and the findings of this manuscript, the authors propose a model, according to which splenic macrophages eliminate intact senescent RBCs, liver macrophages clear RBC ghosts, while LSECs take up and catabolize released Hb. The overall data are of high technical quality and most experiments are appropriately controlled. There are, however, issues that require attention.

- 1) It is not clear to which extent LSECs contribute to iron recycling under physiological conditions. Is this just a stress-response mechanism, or are LSECs key regulators of physiological heme iron recycling? Which is the relative percentage of heme that is recycled by LSECs, splenic macrophages and Kupffer cells under physiological and stress conditions?
- 2) The data with BALB/c and BL/6J mice appear overinterpreted. The difference in erythrophagocytic capacity of splenic macrophages is attributed to different ferroportin expression. However, these parameters are not directly connected. Is Hmox1 expression in splenic macrophages and LSECs different among the two strains? Can the iron phenotype be reversed with injection of PR73 mini-hepcidin?
- 3) CD163 knock-out mice do not show major differences in iron metabolism under physiological conditions. Is this also the case after phenylhydrazine-induced hemolysis?
- 4) According to Fig. 6E, phenylhydrazine mainly induces Hmox1 mRNA in LSECs vs Kupffer cells. This should be validated by immunohistochemistry.
- 5) Figure 2I: Is albumin uptake also blocked by EIPA and partially blocked by LAT-A, ML141 and PRI-724?
- 6) In Figure 2B, why did LSECs take up free Hb more robustly than haptoglobin-bound Hb? Please, comment.
- 7) In the section describing data with Ubi-GFP transgenic mice, please briefly explain the model and provide a reference.
- 8) The manuscript would benefit by adding a paragraph describing limitations of the study.
- 9) The manuscript would also benefit by adding a schematic figure with the proposed model.

Referee #3:

In this paper, the authors provide compelling evidence that liver sinusoidal endothelial cells (LSECs) play a major role in recycling free hemoglobin in the blood via macropinocytosis, a situation that arises in conditions such as transfusion, malaria, sickle cell disease, and thalassemia. By contrast, red pulp macrophages primarily recycle intact senescent red blood cells, while liver Kupffer cells can recycle red blood cell ghosts and debris. Compared to LSECs, splenic and aortic endothelial cells play a minimal role in hemoglobin recycling. Consistently, LSECs exhibit a unique transcriptomic and proteomic signature indicative of heme catabolism, ferritin iron storage, ferroptosis, and oxidative response, among other pathways. This LSEC role is new and surprising, but well supported in the ms.

Questions and suggestions:

What was the specific reason or concern behind using female mice or primary female LSECs in this study?

Line 193, Error: CD23B" should be corrected to "CD32B."

Figure 2F, CD36 is annotated in white, but there is no clear white color overlapping with dextran. If the authors intended to represent CD36 with blue staining, it should be labeled blue instead of white.

Additionally, this figure is confusing, especially when viewed alongside Figure 2E, where the blue staining corresponds to Hoechst (nuclear staining). A clearer annotation would improve readability.

Figure S2B - Panel Labeling Issue: The bottom panel is a zoomed-in view of the region inside the square in the top panel. The four panels represent staining in white, red, green, and red/green overlap, apparently. Since the figure contains four different colors (white, blue, red, and green), it would be helpful to label each panel explicitly. Otherwise, by default, the fourth panel is assumed to be a blue channel rather than an overlap.

Figure 2G vs. 2H - EIPA Treatment Effect, the EIPA treatment appears to have a significantly weaker effect on human LSECs compared to mouse LSECs. Could the authors provide an explanation for this difference?

Figure 3 - Hemoglobin Administration Method. In lines 233-250, how was hemoglobin administered to the mice-intravenously or intraperitoneally?

Figure S3A -Please label the x-axis for clarity.

Figure S4B - Lack of Annotations: The figure is confusing and needs more annotation. What do the different grouped cells in the first panel represent? While the last three panels clearly show the expression of SLC40A1, HMOX1, and HMOX2, the first and second panels need additional information for proper interpretation.

Have the authors measured serum haptoglobin and hemopexin levels in CD163 knockout mice? If so, reporting these data would provide valuable insights.

Figures 5-6 - Y-Axis Labeling Issue: Please correct the formatting of "micro" on the y-axis in some of the figures.

Warsaw 08.10.2025

Dear Dr Rembold,

Dear Reviewers,

We sincerely thank the reviewers for their thorough evaluation of our manuscript, their constructive suggestions, and their recognition of the importance of our study. We carefully considered each point raised and, in the revised version, we have either provided additional data, clarified our reasoning, or expanded the interpretation where appropriate. Below, we present a detailed, point-by-point response to all comments.

Reviewer #1:

This is a very extensively documented, comprehensive analysis of the role of liver sinusoidal endothelial cells (LSEC) in the catabolism of hemoglobin generated during hemolysis. The manuscript successfully challenges the existing paradigm that attributes the erythrocyte-degrading function exclusively to macrophages. The authors convincingly demonstrate a novel pathway for the recycling of material from damaged erythrocytes wherein the spleen releases free hemoglobin and erythrocyte ghosts to the liver. In the liver, LSECs take up the free hemoglobin, and Kupffer cells (macrophages) take up the red cell ghosts. The authors extensively document the specialization of LSECs in hemoglobin catabolism and their dominant role in handling free hemoglobin releases by hemolysis. They also show the effect of hemoglobin uptake by LSECs on iron regulation. Hemoglobin uptake induces secretion of the critical iron regulator BMP6 from LSECs. BMP6 then induces hepcidin secretion from hepatocytes, allowing hemoglobin produced by hemolysis to influence systemic iron metabolism. Overall, this is a tour de force contribution to our understanding of the coordinated response of the organism to hemolysis, with implications for a variety of important pathophysiological settings including hemolytic anemias and transfusion therapy, and possibly also normal erythrocyte turnover.

This will be an influential contribution to the field of erythrocyte biology and iron metabolism.

My suggestions are minor:

1) Lines 120-125 should be reworded: "A sterile aqueous iron citrate (FeCit, 150 µg/mouse) solution (Sigma-Aldrich, #F3388) or sterile citric acid buffer (0.05 M, Sigma-Aldrich, #251275) was normalized to pH 7.0 and administered i.v. for 5 h. Mini-hepcidin (PR73, 50 nmol/mouse) (kind gift from Elizabeta Nemeth, UCLA, USA) was injected intraperitoneally (i.p.) for 4 h. To induce hemolysis, a sterile solution of phenylhydrazine (PHZ) (Sigma-Aldrich, #P26252) in PBS was administered i.p. at a dose of 0.125 mg/g of body weight for 6 h."

Reword to indicate that tissues were harvest X number of hours after the administration of substances, rather than imply that the substances were administered for X hours.

Ad R1.1. We thank the reviewer for highlighting the ambiguity. We have revised the Materials and Methods section to specify that tissue collection took place at a defined number of hours after administration, rather than during ongoing administration. This is now included in lines 432-439.

2) Figure 4H and line 286: The authors do not statistically analyze the difference between the responses of LSEC and KC to PR73 so they cannot make comparisons other than that the responses appear similar.

Ad.R1.2. We agree that our conclusions were somewhat overstated. We have revised the indicated sentence to more accurately state that the responses of LSECs and KCs were comparable upon PR73 injection (line 233).

3) Figure 4A, FPN panel: It is hard to see what the arrows are pointing to. The related CD146 and F4/80 panels also appear dim.

AdR1.3 We agree that the micrographs appeared dim. We have now prepared zoomed-in images with enhanced brightness and contrast, adjusted to levels at which the negative (blank) staining remains negative.

Reviewer #2:

This study provides evidence that liver sinusoidal endothelial cells (LSECs) have a major role in clearance of free hemoglobin (Hb) from lysed red blood cells (RBCs). They take up Hb via micropinocytosis and express iron-handling proteins, including the heme-degrading enzyme Hmox1. Previous data suggested that there is some degree of hemolysis during physiological erythrophagocytosis in the spleen. Based on this and the findings of this manuscript, the authors propose a model, according to which splenic macrophages eliminate intact senescent RBCs, liver macrophages clear RBC ghosts, while LSECs take up and catabolize released Hb. The overall data are of high technical quality and most experiments are appropriately controlled. There are, however, issues that require attention.

1) It is not clear to which extent LSECs contribute to iron recycling under physiological conditions. Is this just a stress-response mechanism, or are LSECs key regulators of physiological heme iron recycling? Which is the relative percentage of heme that is recycled by LSECs, splenic macrophages and Kupffer cells under physiological and stress conditions?

Ad. R2.1. We thank the reviewer for this important question, which aligns closely with a central objective of our study. To address this point, we first utilized ICP-OES, a spectroscopic method that quantifies total iron levels in biological samples. We now include data comparing iron levels in FACS-sorted LSECs with those in KCs and other ECs, indicating that the levels in LSECs fall between iron-recycling macrophages and ECs (new Figure 4F). By employing a newly established robust method of magnetic sorting of LSECs, KCs, and RPMs, we next measured heme levels in these populations. These data, presented in Figure 4H, show that the heme content of LSECs is not substantially lower than that of KCs or RPMs. We also included a new Appendix Fig. S3 illustrating the efficiency of our magnetic purification method for LSECs and KCs.

Next, we simulated stress conditions by intravenously injecting free hemoglobin. We found that LSECs rapidly increased their total iron content at the early time point upon hemoglobin administration, whereas such a clear response was not evident in KCs or RPMs. These new data are presented in Figure 6E.

2) The data with BALB/c and BL/6J mice appear overinterpreted. The difference in erythrophagocytic capacity of splenic macrophages is attributed to different ferroportin expression. However, these parameters are not directly connected.

Ad. R2.2a. This connection was previously reported in our article by Slusarczyk, Mandal et al. eLife, showing that iron accumulation in RPMs, a response likely resulting from decreased FPN membrane expression, suppresses erythrophagocytic activity. In addition, our recent preprint – Mandal et. al. <https://doi.org/10.1101/2025.06.13.659526>, provides some new insights directly linking membrane FPN expression with erythrophagocytosis capacity. We opted not to cite this preprint as we expect it to evolve further, but we have modified the argumentation in the revised manuscript; please see lines 276-277.

Is Hmox1 expression in splenic macrophages and LSECs different among the two strains? Can the iron phenotype be reversed with injection of PR73 mini-hepcidin?

Ad. R2.2b. We thank the reviewer for this suggestion. We have now measured HO-1 expression by flow cytometry in LSECs, KCs, and RPMs, in BALB/c and C57BL/6J mice. Consistent with other parameters, we found that BALB/c mice generally have a higher abundance of LSECs, including those positive for HO-1. These data are now presented in the main Figure 5 (for LSECs) and for all three cell types in Figure EV4E and EV4F.

We appreciate the reviewer's suggestion to modulate iron parameters in BALB/c and C57BL/6J mice using PR73. However, we believe such experiments fall outside the scope of the current manuscript. PR73 acts acutely, rapidly reducing FPN levels and transferrin saturation. Given this short-term mode of action, it is unlikely to be suitable for probing iron recycling between the spleen and liver in these two strains. Additionally, we measured hepcidin mRNA expression and found no significant differences between BALB/c and C57BL/6J mice, suggesting that strain-specific differences in RPM FPN expression are not strictly governed by hepcidin levels. Finally, performing such interventions would require additional ethical permits, which were a limiting factor during the revision process. For these reasons, we decided not to pursue this experimental approach.

3) CD163 knock-out mice do not show major differences in iron metabolism under physiological conditions. Is this also the case after phenylhydrazine-induced hemolysis?

Ad. R2.3. We are grateful to the reviewer for this suggestion. Unfortunately, CD163 KO mice are no longer maintained as a live colony in the laboratory of Anders Etzerodt, preventing us from performing experiments with this model. Instead, we investigated the role of macrophages in hemoglobin clearance using a clodronate depletion approach. Furthermore, we chose to transfuse hemoglobin rather than administer PHZ, as this provides a cleaner and more controlled method to mimic excessive free hemoglobin exposure.

Our new data, presented in Figures 5H and 5I, show that macrophage depletion with clodronate does not alter heme accumulation in plasma or the kidneys. These findings indicate that short-term hemoglobin clearance is not substantially impaired by macrophage deficiency.

4) According to Fig. 6E, phenylhydrazine mainly induces Hmox1 mRNA in LSECs vs Kupffer cells. This should be validated by immunohistochemistry.

Ad R2.4. Consistent with our argumentation above (Ad R2.3), we chose to use Hb transfusion rather than PHZ, as it provides a cleaner and more controlled method of mimicking a rapid increase in free Hb levels, and it also represents the primary approach used throughout the study. Using Hb administration, we found that although KCs exhibit higher basal HO-1 expression than LSECs, this difference becomes less pronounced after Hb injection, suggesting a greater contribution of LSECs to HO-1 expression in the liver. These data are now included in Figure 6E.

5) Figure 2I: Is albumin uptake also blocked by EIPA and partially blocked by LAT-A, ML141 and PRI-724?

Ad. R2.5. We thank the reviewer for this question. New Figure 2J shows that primary LSECs efficiently internalize fluorescent albumin, and that this uptake is inhibited by EIPA, LAT-A, and PRI-724.

6) In Figure 2B, why did LSECs take up free Hb more robustly than haptoglobin-bound Hb? Please, comment.

Ad. R26. We have reconsidered our conclusions based on this data. The observed difference in uptake efficiency between free Hb and the Hb:Hp complex may be influenced by the size of cargo; however, a detailed characterization of this aspect was beyond the scope of our study. Preparation of the Hb:Hp complex required additional purification using Amicon columns, while free Hb was processed in parallel using the same procedure. Importantly, we may not have remeasured Hb concentrations for both cargoes after this purification. For this reason, we decided not to perform a direct comparison using an ANOVA test. Instead, we have now stated in the text that both free Hb and Hb:Hp complex appear to be sequestered comparably (line 128).

7) In the section describing data with Ubi-GFP transgenic mice, please briefly explain the model and provide a reference.

Ad. R2.7 The UBI-GFP model is now described in greater detail in the Methods (lines 411-413) and in the Results (lines 246-248) sections. We have also included the exact JAX ID number as a reference, given that these mice are a commonly used strain.

8) The manuscript would benefit by adding a paragraph describing limitations of the study.

Ad. R2.8. This section has been added at the end of the Discussion.

9) The manuscript would also benefit by adding a schematic figure with the proposed model.

Ad R2.9 We have now included the graphical abstract.

Reviewer #3:

In this paper, the authors provide compelling evidence that liver sinusoidal endothelial cells (LSECs) play a major role in recycling free hemoglobin in the blood via macropinocytosis, a situation that arises in conditions such as transfusion, malaria, sickle cell disease, and thalassemia. By contrast, red pulp macrophages primarily recycle intact senescent red blood

cells, while liver Kupffer cells can recycle red blood cell ghosts and debris. Compared to LSECs, splenic and aortic endothelial cells play a minimal role in hemoglobin recycling. Consistently, LSECs exhibit a unique transcriptomic and proteomic signature indicative of heme catabolism, ferritin iron storage, ferroptosis, and oxidative response, among other pathways. This LSEC role is new and surprising, but well supported in the ms. Questions and suggestions:

1. What was the specific reason or concern behind using female mice or primary female LSECs in this study?

Ad. R3.1 To our knowledge, detailed comparative studies of iron recycling from erythrocytes between male and female mice are lacking. Available evidence suggests that females may have less efficient erythrophagocytosis by splenic macrophages, with aging-related loss of RPM function and progressive iron accumulation in the spleen being more pronounced in females (Slusarczyk, Mandal et al. eLife and Altamura et al. Cell Met). Notably, foundational work for our study (Klei et al., "Hemolysis in the spleen drives erythrocyte turnover") also used female mice. Therefore, using female mice in our study ensures experimental consistency and allows direct cross-study comparison. Nevertheless, we expect our findings to be relevant to males as well, as primary human non-parenchymal cells obtained from a male patient also demonstrated Hb uptake. A detailed comparison of the extent to which LSECs contribute to iron recycling in males versus females, however, would require an independent study and lies beyond the scope of the present work. The points above are now mentioned in the manuscript (please see lines 420-424)

2. Line 193, Error: CD23B" should be corrected to "CD32B." Figure 2F, CD36 is annotated in white, but there is no clear white color overlapping with dextran. If the authors intended to represent CD36 with blue staining, it should be labeled blue instead of white.

Ad R3.2. We are grateful to the Reviewer for pointing out these errors; both have now been corrected.

3. Additionally, this figure is confusing, especially when viewed alongside Figure 2E, where the blue staining corresponds to Hoechst (nuclear staining). A clearer annotation would improve readability.

Ad. R3.3. We have now clarified the color panels indicating the stained organelles and proteins.

4. Figure S2B - Panel Labeling Issue: The bottom panel is a zoomed-in view of the region inside the square in the top panel. The four panels represent staining in white, red, green, and red/green overlap, apparently. Since the figure contains four different colors (white, blue, red, and green), it would be helpful to label each panel explicitly. Otherwise, by default, the fourth panel is assumed to be a blue channel rather than an overlap.

Ad. R3. 4. We thank the reviewer for noticing this unclear labeling. We have now indicated specific staining for each panel.

5. Figure 2G vs. 2H - EIPA Treatment Effect, the EIPA treatment appears to have a significantly weaker effect on human LSECs compared to mouse LSECs. Could the authors provide an explanation for this difference?

Ad R3.5. Indeed, EIPA shows a more pronounced inhibition of hemoglobin uptake in mouse primary LSECs compared to human LSECs. This difference likely reflects species-specific variations in the expression and regulation of sodium-hydrogen exchangers targeted by EIPA, which underlie macropinocytosis. Additional factors, including donor variability, in vitro cell adaptation, and differences in membrane composition, may further reduce the effectiveness of EIPA in human LSECs. Nevertheless, both mouse and human LSECs show comparable responses to LAT-A-mediated inhibition of actin remodeling, suggesting that Hb-uptake at least partially depends on macropinocytosis in both species, but not on classical clathrin-dependent endocytosis. Finally, we cannot exclude the contribution of EIPA-independent, compensatory routes of hemoglobin uptake that may operate in human cells.

These points are now mentioned in the respective parts of Results, please see lines 141-143 and 150-152.

6. Figure 3 - Hemoglobin Administration Method. In lines 233-250, how was hemoglobin administered to the mice-intravenously or intraperitoneally?

Ad. R3.6. We thank the Reviewer for noticing this unclear description. In all our in vivo experiments, hemoglobin was delivered intravenously; this is now clarified in the Results and the figure legends.

7. Figure S3A -Please label the x-axis for clarity.

Ad R3.7. We thank the Reviewer for noticing the missing label, which has now been corrected.

8. Figure S4B - Lack of Annotations: The figure is confusing and needs more annotation. What do the different grouped cells in the first panel represent? While the last three panels clearly show the expression of SLC40A1, HMOX1, and HMOX2, the first and second panels need additional information for proper interpretation.

Ad R3.8. We agree that the figure legend was previously unclear. We have now expanded the explanation of the panels. The expression data are shown not only for LSECs but also for other endothelial cells in the liver, reflecting our intention to capture all potential signals of SLC40A1, HMOX1, and HMOX2 in porcine liver endothelium, rather than restricting the analysis to cells labeled specifically as porcine LSECs.

9. Have the authors measured serum haptoglobin and hemopexin levels in CD163 knockout mice? If so, reporting these data would provide valuable insights.

Ad R3.9. Unfortunately, these analyses were not performed, and no sufficient amount of plasma was left after other measurements to address this additional aspect. However, we believe that both hemopexin and haptoglobin are unlikely to be affected, as heme levels were not significantly different between wild-type and CD163 KO mice either in the circulation or the splenic extracellular fluid.

10. Figures 5-6 - Y-Axis Labeling Issue: Please correct the formatting of "micro" on the y-axis in some of the figures.

Ad R3.10. This was clearly a figure conversion error, which has now been corrected in the revised version.

Sincerely,

K. Mleczko-Sanecka

Katarzyna Mleczko-Sanecka, PhD

Head of the Laboratory of Iron Homeostasis,
International Institute of Molecular and Cell Biology,
Warsaw, Poland

Dear Dr. Mleczko-Sanecka

Thank you for the submission of your revised manuscript to EMBO reports. It has been seen again by former referee #2 who considers all concerns adequately addressed and recommends publication.

Browsing through the manuscript myself, I noticed a few editorial things that we need before we can proceed with the official acceptance of your study.

- Please provide up to 5 keywords on the title page.

- Please update the 'Conflict of interest' paragraph to our new 'Disclosure and competing interests statement'. For more information see

<https://www.embopress.org/page/journal/14693178/authorguide#conflictsofinterest>

- Co-author Daniel Seehofer is listed in the online manuscript tracking system as author but not part of the author list on the manuscript. Please check.

- Regarding the Author Contributions, we now use CRediT to specify the contributions of each author in the journal submission system. Therefore, please remove the Author Contributions from the manuscript file and make sure that the author contributions in our online manuscript tracking system are correct and up-to-date. The information you specified in the system will be automatically retrieved and typeset into the article. You can enter additional information in the free text box provided, if you wish. See also our guide to authors <https://www.embopress.org/page/journal/14693178/authorguide#authorshipguidelines>.

- References: et al needs to be used after the 10th author name.

- The funding information in the paper needs to be congruent with that in the online manuscript tracking system. In fact, the information in the database is the one that matters for deposition to PubMed. It appears like these are missing in the online system: European Union under the European Regional Development Fund; the European Union-NextGenerationEU under the National Recovery and Resilience Plan; the European Union under the European Funds for Smart Economy 2021-2027 (FENG).

- The use of BioRender should be acknowledged in the methods section (only) as follows:

Graphics:

(some of the... OR Figure #... OR synopsis) Graphics were created with BioRender.com.

- The scale bar size should not be defined in the image itself (e.g., as in Fig 1F, 2E, 4B, EV1C), only in the legend.

- The scale bars in Figure 4A are very thin and hard to see.

- Dataset EV1: Please add a short legend or description and the title 'Dataset EV1' on the title page of the file. The file should be uploaded as file type 'Dataset'.

- For Datasets EV2 and EV3: please add a legend in a separate tab of the .xls file.

- The Appendix needs a title page with a table of content and page numbers.

- The resolution of Appendix Fig. S2B, C appears rather low.

- Please provide the specific URLs for GSE235976, GSE240270, PXD051274 datasets in the data availability statement. I.e., we need links that resolve directly to the datasets. Please also provide information and a link to the original FCS files you planned to upload to BioStudies.

- Please address the following points noted by our data editors in the figure legends:

1) The legend for figure 3 needs to be provided in a sequential manner. Currently, it lists (C) after (E) and (G) etc.

2) Please provide information related to n in the legends of figures 1A, B; 2A, B, G, I, J; 3A, B, C, D, E, F, G, H; 4C, D, F, G, H, I, J; 5A-N; 6A, B, C, D, F, G, H, I; 7A-F; EV1 A, D; EV2 D, E, F, G; EV3 A-F; EV4 A-F; EV5 A-D.

3) The data editors noted that the exact p-values were not provided in the legends of figures 1A, B; 2G, I, J; 3C, D, F, G; 4C, F, I; 5A, B, D, H; 6C, D; 7A-C; EV1 A, EV2 D, E, F, G; EV5 B.

I spot-checked a few and noticed that the exact p-values were actually specified but in the figure panels. If this is true for all of the listed panels, it is fine. Just make sure please that all exact p-values are there.

- We perform a routine image integrity check on all revised manuscripts. In this case we noticed two instances of image reuse between different figures as follows:

1) Cell reuse between Figure 1C and Figure EV1C. It looks as if EV1C shows the single channel images for Fig. 1C, which is

perfectly fine but this re-use must be clearly stated in both figure legends, please.

2) The same applies to the cell reuse between Figure 2C & D and Figure EV2A & B.

- To maintain consistency and avoid any possible misinterpretation of data, I kindly ask that you upload the microscopy source data for Figures EV1 and EV2 & Appendix Fig S2. This will allow us to confirm the integrity of the full figure set and support transparency for readers.

- Please provide a short summary that will accompany the synopsis image online as follows:

A) a short (1-2 sentences) summary of the findings and their significance,

B) 2-3 bullet points highlighting key results.

- Finally, please describe all new findings in the abstract in present tense.

With kind regards,

=====

Referee #2:

The revised manuscript is improved and has addressed all issues raised by the reviewers

The authors have addressed all minor editorial requests.

Dr. Katarzyna Mleczko-Sanecka
International Institute of Molecular and Cell Biology in Warsaw
Laboratory of Iron Homeostasis
4 Ks. Trojdena Street
Warsaw 02-109
Poland

Dear Katarzyna,

Thank you for your patience while we have performed our final editorial checks. I am now very pleased to accept your manuscript for publication in the next available issue of EMBO reports. Thank you for your contribution to our journal.

You may qualify for financial assistance for your publication charges - either via a Springer Nature fully open access agreement or an EMBO initiative. Check your eligibility: <https://link.springer.com/journal/44319/how-to-publish-with-us>

Kind regards,

Martina

>>> Please note that it is EMBO Reports policy for the transcript of the editorial process (containing referee reports and your response letter) to be published as an online supplement to each paper. If you do NOT want this, you will need to inform the Editorial Office via email immediately. More information is available here: <https://link.springer.com/partners/embo-press/editorial-policies#Peer%20review>